# A Gradient Method for Multilevel Optimization

**Ryo Sato**
The University of Tokyo

**Mirai Tanaka**
The Institute of Statistical Mathematics
RIKEN

**Akiko Takeda**
The University of Tokyo
RIKEN

## Abstract

Although application examples of multilevel optimization have already been discussed since the 1990s, the development of solution methods was almost limited to bilevel cases due to the difficulty of the problem. In recent years, in machine learning, Franceschi et al. have proposed a method for solving bilevel optimization problems by replacing their lower-level problems with the $T$ steepest descent update equations with some prechosen iteration number $T$. In this paper, we have developed a gradient-based algorithm for multilevel optimization with $n$ levels based on their idea and proved that our reformulation asymptotically converges to the original multilevel problem. As far as we know, this is one of the first algorithms with some theoretical guarantee for multilevel optimization. Numerical experiments show that a trilevel hyperparameter learning model considering data poisoning produces more stable prediction results than an existing bilevel hyperparameter learning model in noisy data settings.

## 1 Introduction

**Multilevel optimization** When modeling real-world problems, there is often a hierarchy of decision-makers, and decisions are taken at different levels in the hierarchy. In this work, we consider multilevel optimization problems whose simplest form[1] is the following:

$$
\begin{aligned}
\min_{x_1 \in S_1 \subseteq \mathbb{R}^{d_1}, x_2^*, \ldots, x_n^*} & \; f_1(x_1, x_2^*, \ldots, x_n^*) \text{ s.t.} \\
x_2^* = & \operatorname*{argmin}_{x_2 \in \mathbb{R}^{d_2}, x_3^*, \ldots, x_n^*} f_2(x_1, x_2, x_3^*, \ldots, x_n^*) \text{ s.t.} \\
& \qquad \ddots \\
& \; x_n^* = \operatorname*{argmin}_{x_n \in \mathbb{R}^{d_n}} f_n(x_1, x_2, \ldots, x_n),
\end{aligned}
\tag{1}
$$

where $d_i$ is positive integers, $x_i \in \mathbb{R}^{d_i}$ is the decision variable at the $i$th level, and $f_i \colon \mathbb{R}^{d_i} \to \mathbb{R}$ is the objective function of the $i$th level for $i = 1, \ldots, n$. In the formulation, an optimization problem contains another optimization problem as a constraint. The framework can be used to formulate problems in which decisions are made in sequence and earlier decisions influence later decisions. An optimization problem in which this structure is $n$-fold is called an $n$-level optimization problem or a multilevel optimization problem especially when $n \geq 3$.

**Existing study on bilevel optimization** Especially, when $n = 2$ in (1), it is called a bilevel optimization problem. Bilevel optimization or multilevel optimization has been a well-known problem in the field of optimization since the 1980s or 1990s respectively (see the survey paper [24] for the research at that time), and their solution methods have been studied mainly on bilevel

---

[1]This assumes that lower-level problems have unique optimal solutions $x_2^*, \ldots, x_n^*$ for the simplicity.

35th Conference on Neural Information Processing Systems (NeurIPS 2021).

optimization until now. Recently, studies on bilevel optimization have received significant attention from the machine learning community due to practical applications and the development of efficient algorithms. For example, bilevel optimization was used to formulate decision-making problems for players in conflict [25, 22], and to formulate hyperparameter learning problems in machine learning [5, 11, 12]. Franceschi et al. [11] constructed a single-level problem that approximates a bilevel problem by replacing the lower-level problem with $T$ steepest descent update equations using a prechosen iteration number $T$. It is shown that the optimization problem with $T$ steepest descent update equations asymptotically converges to the original bilevel problem as $T \to \infty$. It is quite different from the ordinary approach (e.g., [1]) that transforms bilevel optimization problems into single-level problems using the optimality condition of the lower-level problem. The application of the steepest descent method to bilevel optimization problems has recently attracted attention, resulting in a stream of papers; e.g., [13, 20, 18].

**Existing study on multilevel optimization**    Little research has been done on decision-making models (1) that assume the multilevel hierarchy, though various applications of multilevel optimization have been already discussed in the 1990s [24]. As far as we know, few solution methods with a theoretical guarantee have been proposed. A metaheuristic algorithm based on a perturbation, i.e., a solution method in which the neighborhood is randomly searched for a better solution in the order of $x_1$ to $x_n$ has been proposed in [23] for general $n$-level optimization problems. The effect of updating $x_i$ on the decision variables $x_{i+1}, \ldots, x_n$ of the lower-level problems is not considered, and hence, the hierarchical structure of the multilevel optimization problem cannot be fully utilized in updating the solution. Because of this, the algorithm does not have a theoretical guarantee for the obtained solution. Very recently (two days before the NeurIPS 2021 abstract deadline), a proximal gradient method based on the fixed-point theory for trilevel optimization problem [21] has been proposed. This paper has proved convergence to an optimal solution assuming the convexity of the objective functions. However, it does not include numerical results and hence its practical efficiency is unclear.

**Contribution of this paper**    In this paper, by extending the gradient method for bilevel optimization problems [11] to multilevel optimization problems, we propose an algorithm with a convergence guarantee for multilevel optimization other than one by [21]. Increasing the problem hierarchy from two to three drastically makes developing algorithms and showing theoretical guarantees difficult. In fact, as discussed above, in the case of bilevel, replacing the second-level optimization problem with its optimality condition or replacing it with $T$ steepest-descent sequential updates immediately results in a one-level optimization problem. However, when it comes to $n \geq 3$ levels, it may be necessary to perform this replacement $n$ times, and it seems not easy to construct a solution method with a convergence guarantee. It is not straightforward at all to give our method a theoretical guarantee similar to the one [11] developed for bilevel optimization. We have confirmed the effectiveness of the proposed method by applying it to hyperparameter learning with real data. Experimental verifications of the effectiveness of trilevel optimization using real-world problems are the first ones as far as we know.

## 2    Related existing methods

### 2.1    Two-stage robust optimization problems

When making long-term decisions, it is also necessary to make many decisions according to the situation. If the same objective function is acceptable throughout the period, i.e., $f_1 = \cdots = f_n$, and there are hostile players, such a problem is often formulated as a multistage robust optimization problem. Especially, the concept of two-stage robust optimization (so-called adjustable robust optimization) $\min_{x_1 \in S_1} \max_{x_2 \in S_2} \min_{x_3 \in S_3} f_1(x_1, x_2, x_3)$ was introduced with feasible sets $S_i \subseteq \mathbb{R}^{d_i}$ for $i = 1, 2, 3$, and methodology based on affine decision rules was developed by [4]. Researches based on affine policies are still being actively conducted; see, e.g., [6, 26]. The two-stage robust optimization can be considered as a special case of (1) with $n = 3$ when constraints $x_2 \in S_2 \subseteq \mathbb{R}^{d_2}$ and $x_3 \in S_3 \subseteq \mathbb{R}^{d_3}$ are added. This approach is unlikely to be applicable to our problem (1) without strong assumptions such as affine decision rules.

There is another research stream on min-max-min problems including integer constraints (see e.g., [7, 25]). They transform the problem into the min-max problem by taking the dual for the inner "min" and apply cut-generating approaches or Benders decomposition techniques with the property of

integer variables. While the approach is popular in the application studies of electric grid planning, it is restricted to the specific min-max-min problems and no more applicable to multilevel optimization problems (1).

## 2.2 Existing methods for bilevel optimization problems

Various applications are known for bilevel optimization problems, i.e., the case of $n = 2$ in (1):

$$\min_{x_1 \in S_1} f_1(x_1, x_2) \text{ s.t. } x_2 = \operatorname*{argmin}_{x_2'} f_2(x_1, x_2'). \tag{2}$$

One of the most well-known applications in machine learning is hyperparameter learning. For example, the problem that finds the best hyperparameter value in the ridge regression is formulated as

$$\min_{\lambda \geq 0} \|y_{\text{valid}} - X_{\text{valid}}\theta\|_2^2 \text{ s.t. } \theta = \operatorname*{argmin}_{\theta'} \|y_{\text{train}} - X_{\text{train}}\theta'\|_2^2 + \lambda\|\theta'\|_2^2,$$

where $(X_{\text{train}}, y_{\text{train}})$ are training samples, $(X_{\text{valid}}, y_{\text{valid}})$ are validation samples, and $\lambda$ is a hyperparameter that is optimized in this problem. Under Assumption 1 restricted to $n = 2$ shown later, the existence of an optimal solution to the bilevel problem (2) is ensured by [12, Theorem 3.1].

There are mainly two approaches to solve bilevel optimization problems. The old practice is to replace the lower-level problem with its optimality condition and to solve the resulting single-level problem. It seems difficult to use this approach to multilevel problems because we need to apply the replacement $n$-times. The other one is to use gradient-based methods developed by Franceschi et al. [11, 12] for bilevel optimization problems. Their approach reduces (2) to a single-level problem by replacing the lower-level problem with $T$ equations using a prechosen number $T$.

Hereinafter, we briefly summarize the results of Franceschi et al. [11, 12] for bilevel optimization. Under the continuous differentiability assumption for $f_2$, $x_2$ is iteratively updated by $x_2^{(t)} = \Phi^{(t)}(x_1, x_2^{(t-1)})$ at the $t$th iteration using an iterative method, e.g., the gradient descent method:

$$\Phi^{(t)}(x_1, x_2^{(t-1)}) = x_2^{(t-1)} - \alpha^{(t)} \nabla_{x_2} f_2(x_1, x_2^{(t-1)}).$$

Then, the bilevel optimization problem is approximated by the single-optimization problem with $T$ equality constraints and new variables $\{x_2^{(t)}\}_{t=1}^T$ instead of $x_2$:

$$\min_{x_1 \in S_1, \{x_2^{(t)}\}} f_1(x_1, x_2^{(T)}) \text{ s.t. } x_2^{(t)} = \Phi^{(t)}(x_1, x_2^{(t-1)}) \ (t = 1, \ldots, T), \tag{3}$$

where $x_2^{(0)}$ is a given constant. Eliminating $x_2^{(1)}, \ldots, x_2^{(T)}$ using constraints, we can equivalently recast the problem above into the unconstrained problem $\min_{x_1 \in S_1} \tilde{F}_1(x_1)$.

Under assumption on differentiability, the gradient of $\tilde{F}_1(x_1)$ is given in [11, Section 3] by

$$\nabla_{x_1} \tilde{F}_1(x_1) = \nabla_{x_1} f_1(x_1, x_2^{(T)}) + \sum_{t=1}^T B^{(t)} \left( \prod_{s=t+1}^T A^{(s)} \right) \nabla_{x_2} f_1(x_1, x_2^{(T)}),$$

$$A^{(t)} = \nabla_{x_2} \Phi^{(t)}(x_1, x_2^{(t-1)}) \ (t = 1, \ldots, T),$$

$$B^{(t)} = \nabla_{x_1} \Phi^{(t)}(x_1, x_2^{(t-1)}) \ (t = 1, \ldots, T).$$

Roughly speaking, [12, Theorem 3.2] proved that the optimal value and the solution set of (3) converge to those of (2) as $T \to \infty$ under Assumptions 1 and 3 restricted to $n = 2$.

## 3 Multilevel optimization problems and their approximation

We will develop an algorithm for multilevel optimization problems (1) under some assumptions by extending the studies [11, 12] on bilevel problems. Our algorithm and its theoretical guarantee look similar to those in [11, 12], but they are not straightforwardly obtained. As emphasized in Section 1, unlike the bilevel problem, even if the lower-level problem is replaced with $T$ steepest-descent sequential updates, the resulting problem is still a multilevel problem. In this section, we show how to resolve these difficulties that come from the multilevel hierarchy.

### 3.1 Existence of optima of multilevel optimization problems

First, we discuss the existence of an optimal solution of the multilevel optimization problem (1). To do so, we introduce the following assumption, which is a natural extension of that in [12].

**Assumption 1.**

(i) $S_1$ is compact.

(ii) For $i = 1, \ldots, n$, $f_i$ is jointly continuous.

(iii) For $i = 2, \ldots, n$, the set of optimal solutions of the $i$th level problem with arbitrarily fixed $(x_1, \ldots, x_{i-1})$ is a singleton.

(iv) For $i = 2, \ldots, n$, the optimal solution of the $i$th level problem remains bounded as $x_1$ varies in $S_1$.

Assumption 1-(iii) means that the $i$th level problem with parameter $(x_1, \ldots, x_{i-1})$ has a unique optimizer, that is described as $x_i^*$ in (1) though it should be formally written as $x_i^*(x_1, \ldots, x_{i-1})$ since it is determined by $x_1, \ldots, x_{i-1}$. We define $F_i(x_1, \ldots, x_i) := f_i(x_1, \ldots, x_i, x_{i+1}^*, \ldots, x_n^*)$ by eliminating $x_{i+1}^*, \ldots, x_n^*$ since it depends only on $(x_1, \ldots, x_i)$. Then, the $i$th level problem with parameter $(x_1, \ldots, x_{i-1})$ can be written as $\min_{x_i} F_i(x_1, \ldots, x_i)$. Similarly, in the 1st level problem, for fixed $x_1$, the remaining variables $x_2, x_3, \ldots, x_n$ can be represented as $x_2^*(x_1), x_3^*(x_1, x_2^*(x_1))$ and so on. In what follows, we denote them by $x_2^*(x_1), x_3^*(x_1), \ldots, x_n^*(x_1)$ since they are determined by $x_1$. Eliminating them, we can rewrite the 1st level problem, equivalently (1), as

$$\min_{x_1 \in S_1} F_1(x_1). \tag{4}$$

**Theorem 2.** *Under Assumption 1, Problem* (4) *admits optimal solutions.*

Theorem 2 is a generalization of [12, Theorem 3.1], which is valid only for $n = 2$, to general $n$. It is difficult to extend the original proof to general $n$ because, to derive the continuity of $x_i^*$ extending the original proof, a sequence $(x_1, \ldots, x_{i-1})$ approaches to its accumulation point only from a specific direction. Instead, in the multilevel case, we employ the theory of point-to-set mapping. A proof of Theorem 2 is shown in Supplementary material A.1.

### 3.2 Approximation by iterative methods

We extend the gradient method for bilevel optimization problems proposed by Franceschi et al. [11] to multilevel optimization problems (1). Based on an argument similar to one in Subsection 2.2, we approximate lower level problems in (1) by applying an iterative method with $T_i$ iterations to the $i$th level problem for $i = 2, \ldots, n$. Then, we obtain the following approximated problem:

$$\min_{x_1 \in S_1, \{x_2^{(t_2)}\}, \ldots, \{x_n^{(t_n)}\}} f_1(x_1, x_2^{(T_2)}, \ldots, x_n^{(T_n)})$$

$$\text{s.t. } x_i^{(t_i)} = \Phi_i^{(t_i)}(x_1, x_2^{(T_2)}, \ldots, x_{i-1}^{(T_{i-1})}, x_i^{(t_i-1)}) \ (i = 2, \ldots, n; t_i = 1, \ldots, T_i),$$

$$\tag{5}$$

where $\Phi_i^{(t_i)}(x_1, x_2^{(T_2)}, \ldots, x_{i-1}^{(T_{i-1})}, x_i^{(t_i)})$ is the $t_i$th iteration formula of an iterative method for the $i$th level approximated problem with parameters $(x_1, \ldots, x_{i-1}) = (x_1, x_2^{(T_2)}, \ldots, x_{i-1}^{(T_{i-1})})$, i.e.,

$$\min_{x_i, \{x_{i+1}^{(t_{i+1})}\}, \ldots, \{x_n^{(t_n)}\}} f_i(x_1, \ldots, x_i, x_{i+1}^{(T_{i+1})}, \ldots, x_n^{(T_n)})$$

$$\text{s.t. } x_j^{(t_j)} = \Phi_j^{(t_j)}(x_1, \ldots, x_i, x_{i+1}^{(T_{i+1})}, \ldots, x_{j-1}^{(T_{j-1})}, x_j^{(t_j-1)}) \ (j = i+1, \ldots, n; t_j = 1, \ldots, T_j),$$

and its initial point $x_i^{(0)}$ for each $i$ are regarded as a given constant. For $i = 1, \ldots, n$, if we fix $x_1, \ldots, x_i$, each remaining variable $x_j^{(t)}$ is determined by constraints and thus, we denote it by $x_j^{(t_j)}(x_1, \ldots, x_i)$ for $j = i+1, \ldots, n$. Using this notation, the $i$th level objective function can be written as

$$\tilde{F}_i(x_1, \ldots, x_i)$$
$$:= f_i(x_1, \ldots, x_{i-1}, x_i, x_{i+1}^{(T_{i+1})}(x_1, \ldots, x_i), \ldots, x_n^{(T_n)}(x_1, \ldots, x_{n-1}^{(T_{n-1})}(\cdots(x_{i+1}^{(T_{i+1})}(x_1, \ldots, x_i))\cdots))),$$

where $(x_1, \ldots, x_{i-1})$ acts as a fixed parameter in the $i$th level problem. When we approximate the $i$th level problem with the steepest descent method for example, we use $\Phi_i^{(t_i)}(x_1, \ldots, x_{i-1}, x_i^{(t_i)}) = x_i^{(t_i-1)} - \alpha_i^{(t_i-1)} \nabla_{x_i} \tilde{F}_i(x_1, \ldots, x_{i-1}, x_i^{(t_i-1)})$.

Especially, all variables other than $x_1$ in (5) can be expressed as $\{x_2^{(t_2)}(x_1)\}, \ldots, \{x_n^{(t_n)}(x_1)\}$ since it is determined by $x_1$. By plugging the constraints into the objective function and eliminating them, we can reformulate (5) as

$$\min_{x_1 \in S_1} \tilde{F}_1(x_1). \tag{6}$$

In the remainder of this subsection, we assume $T_2 = \cdots = T_n = T$ for simplicity. Then, the optimal value and solutions of Problem (6) converge to those of Problem (4) as $T \to \infty$ in some sense. To derive it, we introduce the following assumption, which is also a natural extension of that in [12].

**Assumption 3.**

(i) $f_1(x_1, \cdot, \ldots, \cdot)$ is uniformly Lipschitz continuous on $S_1$.

(ii) For all $i = 2, \ldots, n$, sequence $\{x_i^{(T)}(x_1)\}$ converges uniformly to $x_i^*(x_1)$ on $S_1$ as $T \to \infty$.

**Theorem 4.** *Under Assumptions 1 and 3, the followings hold:*

*(a) The optimal value of Problem (6) converges to that of Problem (4) as $T \to \infty$.*

*(b) The set of the optimal solutions of Problem (6) converges to that of Problem (4); more precisely, denoting an optimal solution of Problem (6) by $x_{1,T}^*$, we have*

- $\{x_{1,T}^*\}_{T=1}^\infty$ *admits a convergent subsequence;*
- *for every subsequence $\{x_{1,T_k}^*\}_{k=1}^\infty$ such that $x_{1,T_k}^* \to x_1^*$ as $k \to \infty$, the accumulation point $x_1^*$ is an optimal solution of Problem (4).*

See Supplementary material A.2 for a proof of this theorem.

# 4 Proposed method: Gradient computation in the approximated problem

Now we propose to apply a projected gradient method to the approximated problem (6) for multilevel optimization problems (1) because (6) asymptotically converges to (1) as shown in Theorem 4. In this section, we derive the formula of $\nabla_{x_1} \tilde{F}_1(x_1)$ and confirm the local and global convergence of the resulting projected gradient method.

## 4.1 Gradient of the objective function in the approximated problem

The following theorem provides a computation formula of $\nabla_{x_1} \tilde{F}_1(x_1)$.

**Theorem 5** (Gradient formula for the $n$-level optimization problems). *The gradient $\nabla_{x_1} \tilde{F}_1(x_1)$ can be expressed as follows:*

$$\nabla_{x_1} \tilde{F}_1(x_1) = \nabla_{x_1} f_1(x_1, x_2^{(T_2)}, \ldots, x_n^{(T_n)}) + \sum_{i=2}^n Z_i \nabla_{x_i} f_1(x_1, x_2^{(T_2)}, \ldots, x_n^{(T_n)}),$$

$$Z_i = \sum_{t=1}^{T_i} \left( \sum_{j=2}^{i-1} Z_j C_{ij}^{(t)} + B_i^{(t)} \right) \prod_{s=t+1}^{T_i} A_i^{(s)},$$

$$A_i^{(t)} = \nabla_{x_i} \Phi_i^{(t)}(x_1, x_2^{(T_2)}, \ldots, x_{i-1}^{(T_{i-1})}, x_i^{(t-1)}),$$

$$B_i^{(t)} = \nabla_{x_1} \Phi_i^{(t)}(x_1, x_2^{(T_2)}, \ldots, x_{i-1}^{(T_{i-1})}, x_i^{(t-1)}),$$

$$C_{ij}^{(t)} = \nabla_{x_j} \Phi_i^{(t)}(x_1, x_2^{(T_2)}, \ldots, x_{i-1}^{(T_{i-1})}, x_i^{(t-1)})$$

*for any $i = 2, \ldots, n$; $t = 1, \ldots, T_i$; and $j = 2, \ldots, i-1$, where we define $\prod_{s=t+1}^{T_i} A_i^{(s)} := A_i^{(t+1)} A_i^{(t+2)} \ldots A_i^{(T_i)}$ for $t < T_i$ and $\prod_{s=T_i+1}^{T_i} A_i^{(s)} = I$.*

See supplementary material A.3 for a proof of this theorem.

We consider computing $\nabla_{x_1} \tilde{F}_1(x_1)$ using Theorem 5. Notice that we can easily compute $Z_2 = \sum_{t=1}^{T_2} B_2^{(t)} \prod_{s=t+1}^{T_2} A_2^{(s)}$. For $i = 3, \ldots, n$, when we have $Z_2, \ldots, Z_{i-1}$, we can compute $Z_i$. We show an algorithm that computes $\nabla_{x_1} \tilde{F}_1(x_1)$ by computing $Z_2, \ldots, Z_n$ in this order in Algorithm 1.

---

**Algorithm 1** Computation of $\nabla_{x_1} \tilde{F}_1(x_1)$

---

**Input:** $x_1$: current value of the 1st level variable. $\{x_i^{(0)}\}_{i=2}^n$: initial values of the lower level iteration.
**Output:** The exact value of $\nabla_{x_1} \tilde{F}_1(x_1)$.
1: $g := (0, \ldots, 0)^\top$.
2: **for** $i := 2, \ldots, n$ **do**
3:     $Z_i := O$.
4:     **for** $t := 1, \ldots, T_i$ **do**
5:         $x_i^{(t)} := \Phi_i^{(t)}(x_1, x_2^{(T_2)}, \ldots, x_{i-1}^{(T_{i-1})}, x_i^{(t-1)})$.
6:         $\bar{B}_i^{(t)} := \sum_{l=2}^{i-1} Z_l C_{il}^{(t)} + B_i^{(t)}$.
7:         $Z_i := Z_i A_i^{(t)} + \bar{B}_i^{(t)}$.
8: **for** $i = 2, \ldots, n$ **do**
9:     $g := g + Z_i \nabla_{x_i} f_1$.
10: $g := g + \nabla_{x_1} f_1$.
11: **return** $g$

---

For $i = 2, \ldots, n$ and $t = 1, \ldots, T_i$, $\Phi_i^{(t)}$, which appears in the 5th line of Algorithm 1, is the update formula based on the gradient $\nabla_{x_i} \tilde{F}_i(x_1, \ldots, x_i)$ of the $i$th level objective function. $\nabla_{x_i} \tilde{F}_i(x_1, \ldots, x_i)$ can be computed by applying Algorithm 1 to the $(n - i + 1)$-level optimization problem with objective functions $\tilde{F}_i, \ldots, \tilde{F}_n$. Therefore, recursively calling Algorithm 1 in the computation of $\Phi_i^{(t)}$, we can compute $\nabla_{x_1} \tilde{F}_1(x_1)$. For an example of applying Algorithm 1 to Problem (6) arising from a trilevel optimization problem, i.e., Problem (1) with $n = 3$, see Supplementary material B.

## 4.2 Complexity of the gradient computation

We analyze the complexity for computing $\nabla_{x_1} \tilde{F}_1(x_1)$ by recursively calling Algorithm 1. In the following theorem, the asymptotic big O notation is denoted by $\mathrm{O}(\cdot)$.

**Theorem 6.** *Let the time and space complexity for computing $\nabla_{x_i} \tilde{F}_i(x_i)$ be $c_i$ and $s_i$, respectively. We use $\Phi_i^{(t_i)}$ based on $\nabla_{x_i} \tilde{F}_i(x_i)$ and recursively call Algorithm 1 for computing $\nabla_{x_i} \tilde{F}_i(x_1, \ldots, x_i)$. In addition, we assume the followings:*

- *The time and space complexity for evaluating $\Phi_i^{(t_i)}$ are $\mathrm{O}(c_i)$ and $\mathrm{O}(s_i)$.*

- *The time and space complexity of $\nabla_{x_1} f_1$ and $\nabla_{x_i} f_1$ for $i = 1, \ldots, n$ are smaller in the sense of the order than those of $\mathit{for}$ loops in lines 2–9 in Algorithm 1.*

*Then, the overall time complexity $c_1$ and space complexity $s_1$ for computing $\nabla_{x_i} \tilde{F}_i(x_1, \ldots, x_i)$ can be written as*

$$c_1 = \mathrm{O}\left(p^n n! c_n \prod_{i=1}^{n-1} (T_{i+1} d_i)\right), \quad s_1 = \mathrm{O}(q^n s_n), \tag{7}$$

*respectively, for some constant $p, q > 1$.*

For a proof, see Supplementary material A.4. Note that, if $n$ is a fixed parameter, those complexity reduces to a polynomial of $T_i$'s, $d_i$'s, $c_n$, and $s_n$. Hence, Algorithm 1 can be regarded as a fixed-parameter tractable algorithm.

### 4.3 Global convergence of the projected gradient method

Here, we consider solving Problem (6) by the projected gradient method, which calculates the gradient vector by Algorithm 1 and projects the updated point on $S_1$ in each iteration. When all lower-level updates are based on the steepest descent method, we can derive the Lipschitz continuity of the gradient of the objective function of (6). Hence, we can guarantee the local and global convergence of the projected gradient method for (6) by taking a sufficiently small step size.

**Theorem 7.** *Suppose* $\Phi_i^{(t)}(x_1, \ldots, x_{i-1}, x_i^{(t-1)}) = x_i^{(t-1)} - \alpha_i^{(t-1)} \nabla_{x_i} \tilde{F}_i(x_1, \ldots, x_{i-1}, x_i^{(t-1)})$ *for all* $i = 2, \ldots, n$ *and* $t_i = 1, \ldots, T_i$, *where* $\alpha_i^{(t-1)}$ *and* $x_i^{(0)}$ *are given parameters for all* $i$ *and* $t$. *Assume that* $\nabla_{x_j} f_i$ *is Lipschitz continuous and bounded for all* $i = 1, \ldots, n$ *and* $j = 1, \ldots, n$; *and also* $\nabla_{x_i} \Phi_i^{(t)}$, $\nabla_{x_1} \Phi_i^{(t)}$, *and* $\nabla_{x_j} \Phi_i^{(t)}$ *are Lipschitz continuous and bounded for all* $i = 2, \ldots, n$; $j = 2, \ldots, i - 1$; $t = 1, \ldots, T_i$. *Then,* $\nabla_{x_1} \tilde{F}_1$ *is Lipschitz continuous.*

See Supplementary material A.5 for a proof of this theorem.

**Corollary 8.** *Suppose the same assumption as Theorem 7. Assume* $S_1$ *is a compact convex set. Let L be the Lipschitz constant of* $\nabla_{x_1} \tilde{F}_1$. *Then, a sequence* $\{x_1^{(t)}\}$ *generated by the projected gradient method with sufficiently small constant step size, e.g., smaller than* $2/L$, *for Problem* (6) *from any initial point has a convergent subsequence that converges to a stationary point with convergence rate* $O(1/\sqrt{t})$.

*Proof.* From Theorem 7, the gradient of the objective function of Problem (6) is $L$-Lipschitz continuous. Let $G: \text{int}(\text{dom}(\tilde{F}_1)) \to \mathbb{R}^{d_1}$ be the gradient mapping [3, Definition 10.5] corresponding to $\tilde{F}_1$, the indicator function of $S_1$, and the constant step size $\alpha_1^{(t)}$ with satisfying $0 < \alpha_1^{(t)} < 2/L$ for all $t$. Note that $\|G(x_1)\| = 0$ if and only if $x_1$ is a stationary point of Problem (6) [3, Theorem 10.7]. By applying [3, Theorem 10.15], we obtain $\min_{s=0}^{t} \|G(x_1^{(s)})\| \le O(1/\sqrt{t})$ and $\|G(\bar{x}_1)\| = 0$, where $\bar{x}_1$ is a limit point of $\{x_1^{(t)}\}$. $\qquad\square$

## 5 Numerical experiments

To validate the effectiveness of our proposed method, we conducted numerical experiments on an artificial problem and a hyperparameter optimization problem arising from real data (see Supplementary material C for complete results). In our numerical experiments, we implemented all codes with Python 3.9.2 and JAX 0.2.10 for automatic differentiation and executed them on a computer with 12 cores of Intel Core i7-7800X CPU 3.50 GHz, 64 GB RAM, Ubuntu OS 20.04.2 LTS.

In this section, we used Algorithm 1 to calculate the gradient of $\tilde{F}_i$ in problem 5, and used automatic differentiation to calculate the gradient of $\Phi_i^{(t)}$ in problem 5.

### 5.1 Convergence to the optimal solution

We solved the following trilevel optimization problem with Algorithm 1 to evaluate the performance:

$$\min_{x_1 \in \mathbb{R}^2} f_1(x_1, x_2^*, x_3^*) = \|x_3^* - x_1\|_2^2 + \|x_1\|_2^2 \text{ s.t.}$$
$$x_2^* \in \operatorname*{argmin}_{x_2 \in \mathbb{R}^2} f_2(x_1, x_2, x_3^*) = \|x_2 - x_1\|_2^2 \text{ s.t.} \tag{8}$$
$$x_3^* \in \operatorname*{argmin}_{x_3 \in \mathbb{R}^2} f_3(x_1, x_2, x_3) = \|x_3 - x_2\|_2^2.$$

Clearly, the optimal solution for this problem is $x_1 = x_2 = x_3 = (0, 0)^\top$.

We solved (8) with fixed constant step size and initialization but different $(T_2, T_3)$. For the iterative method in Algorithm 1, we employed the steepest descent method at all levels. We show the transition of the value of the objective functions in Figure 1 and the trajectories of each decision variable in Figure 2. Since updates of $x_3$ is the most inner iteration, the time required to update $x_3$ does not change when $T_2$ or $T_3$ changes. Hence the number of updates of $x_3$ is proportional to the total computational time. Therefore, we can compare the time efficiency of the optimization algorithm by

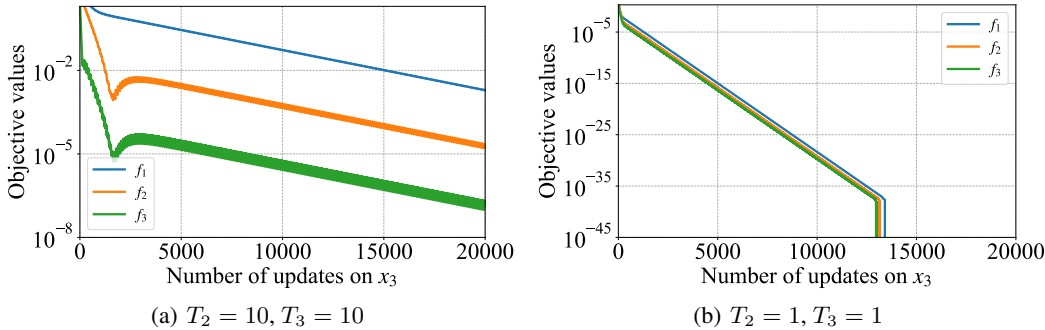

(a) $T_2 = 10, T_3 = 10$            (b) $T_2 = 1, T_3 = 1$

Figure 1: Performance of Algorithm 1 for Problem (8). The objective values linearly decreased.

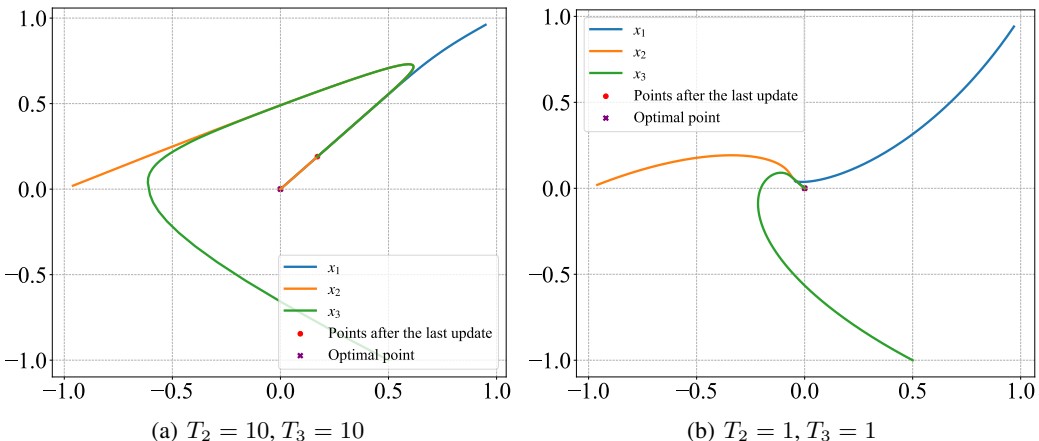

(a) $T_2 = 10, T_3 = 10$            (b) $T_2 = 1, T_3 = 1$

Figure 2: Trajectories of each variable. $(0,0)^\top$ is the optimal solution of each level. Our algorithm with few iterations for lower-level problems $T_2 = T_3 = 1$ performed well.

focusing on the number of updates of $x_3$. We confirmed that the values of $f_1$, $f_2$, and $f_3$ converge to the optimal value 0 at all levels and the gradient method outputs the optimal solution $(0,0)^\top$ even if we use small iteration numbers $T_2 = 1$ and $T_3 = 1$ for the approximation problem (5).

We also made comparison Algorithm 1 and an existing algorithm [23] based on evolutionary strategy by solving (8) using both algorithms. Algorithm 1 outperformed the existing algorithm. For detail, see Supplementary material C.2.

## 5.2 Application to hyperparameter optimization

For deriving a machine learning model robust to noise in input data, we formulate a trilevel model by assuming two players: a model learner and an attacker. The model learner decides the hyperparameter $\lambda$ to minimize the validation error, while the attacker tries to poison training data so as to make the model less accurate. This model is inspired by bilevel hyperparameter optimization [12] and adversarial learning [16, 17] and formulated as follows:

$$\min_\lambda \frac{1}{m} \|y_{\text{valid}} - f(X_{\text{valid}}; \theta)\|_2^2 \text{ s.t.}$$

$$P \in \operatorname*{argmax}_{P'} \frac{1}{n} \|y_{\text{train}} - f(X_{\text{train}} + P'; \theta)\|_2^2 - \frac{c}{nd} \|P'\|_2^2 \text{ s.t.}$$

$$\theta \in \operatorname*{argmin}_{\theta'} \frac{1}{n} \|y_{\text{train}} - f(X_{\text{train}} + P'; \theta')\|_2^2 + \exp(\lambda) \frac{\|\theta'\|_{1*}}{d},$$

where $f$ denotes the output of a three-layer perceptron which has 3 hidden units, $\theta$ denotes the parameter of the model $f$, $d$ denotes the dimension of $\theta$, $n$ denotes the number of the training data

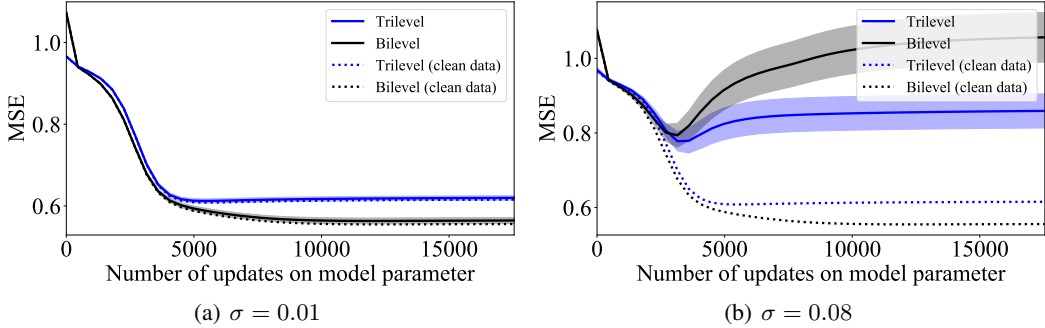

(a) $\sigma = 0.01$  (b) $\sigma = 0.08$

Figure 3: MSE of test data with Gaussian noise with the standard deviation of $\sigma$ on diabetes dataset.

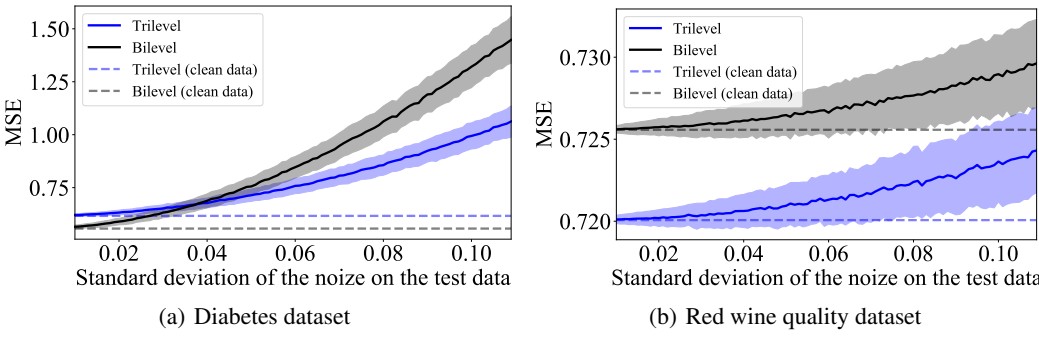

(a) Diabetes dataset  (b) Red wine quality dataset

Figure 4: MSE of test data with Gaussian noise, using early-stopped parameters for prediction.

$X_{\text{train}}$, $m$ denotes the number of the validation data $X_{\text{val}}$, $c$ denotes the penalty for the noise $P$, and $\|\cdot\|_{1*}$ is a smoothed $\ell_1$-norm [19, Eq. (18) with $\mu = 0.25$], which is a differentiable approximation of the $\ell_1$-norm. We used the hyperbolic tangent function as the activation function for the hidden layer of the multilayer perceptron. Here, we use $\exp(\lambda)$ to express a nonnegative penalty parameter instead of the constraint $\lambda \geq 0$.

To validate the effectiveness of our proposed method, we compared the results by the trilevel model with those of the following bilevel model:

$$\min_\lambda \frac{1}{m}\|y_{\text{valid}} - f(X_{\text{valid}}; \theta)\|_2^2 \text{ s.t. } \theta \in \operatorname*{argmin}_{\theta'} \frac{1}{n}\|y_{\text{train}} - f(X_{\text{train}}; \theta')\|_2^2 + \exp(\lambda)\frac{\|\theta'\|_{1*}}{d}.$$

This model is equivalent to the trilevel model without the attacker's level.

We used Algorithm 1 to compute the gradient of the objective function in the trilevel and bilevel models with real datasets. For the iterative method in Algorithm 1, we employed the steepest descent method at all levels. We set $T_2 = 30$ and $T_3 = 3$ for the trilevel model and $T_2 = 30$ for the bilevel model. In each dataset, we used the same initialization and step sizes in the updates of $\lambda$ and $\theta$ in trilevel and bilevel models. We compared these methods on the regression tasks with the following datasets: the diabetes dataset [10], the (red and white) wine quality datasets [8], the Boston dataset [14]. For each dataset, we standardized each feature and the objective variable; randomly chose 40 rows for training data $(X_{\text{train}}, y_{\text{train}})$, chose other 100 rows for validation data $(X_{\text{valid}}, y_{\text{valid}})$, and used the rest of the rows for test data.

We show the transition of the mean squared error (MSE) by test data with Gaussian noise in Figure 3. The solid line and colored belt respectively indicate the mean and the standard deviation over 500 times of generation of Gaussian noise. The dashed line indicates the MSE without noise as a baseline. In the results of the diabetes dataset (Figure 3), the trilevel model provided a more robust parameter than the bilevel model, because the MSE of the trilevel model rises less in the large noise setting for test data.

Next, we compared the quality of the resulting model parameters by the trilevel and bilevel models. We set an early-stopping condition on learning parameters: after 1000 times of updates on the model

Table 1: MSE of test data with Gaussian noise with the standard deviation of 0.08, using early-stopped parameters for prediction. The better values are shown in boldface.

|          | diabetes | Boston | wine (red) | wine (white) |
|----------|----------|--------|------------|--------------|
| Trilevel | **$0.8601 \pm 0.0479$** | **$0.4333 \pm 0.0032$** | **$0.7223 \pm 0.0019$** | **$0.8659 \pm 0.0013$** |
| Bilevel  | $1.0573 \pm 0.0720$ | $0.4899 \pm 0.0033$ | $0.7277 \pm 0.0019$ | $0.8750 \pm 0.0014$ |

parameter, if one time of update on hyperparameter $\lambda$ did not improve test error, terminate the iteration and return the parameters at that time. By using the early-stopped parameters, we show the relationship of test error and standard deviation of the noise on the test data in Figure 4 and Table 1. For the diabetes dataset, the growth of MSE of the trilevel model was slower than that of the bilevel model. For wine quality and Boston house-prices datasets, the MSE of the trilevel model was consistently lower than that of the bilevel model. Therefore, the trilevel model provides more robust parameters than the bilevel model in these settings of problems.

## 5.3   Relationship between $(T_2, T_3)$ and the convergence speed

In the first experiment in Section 5.1, there is no complex relationship between variables at each level, and therefore, the objective function value at one level is not influenced significantly when variables at other levels is changed. In such a problem setting, by setting $T_2$ and $T_3$ to small values, we update $x_1$ many times and the generated sequence $\{x_1^{(t)}\}$ quickly converges to a stationary point. On the other hand, for example, if we set $T_3$ to a large value, $x_3$ is well optimized for some fixed $x_1$ and $x_2$, and hence, our whole algorithm may need more computation time until convergence. In the second experiment in Section 5.2, the relationship between variables at each level is more complicated than in the first experiment. Setting $T_2$ or $T_3$ smaller in such a problem is not necessarily considered to be efficient because the optimization algorithm proceeds without fully approximating the optimality condition of $x_i$ at the $i$th level.

## 6   Conclusion

**Summary**   In this paper, we have provided an approximated formulation for a multilevel optimization problem by iterative methods and discussed its asymptotical properties of it. In addition, we have proposed an algorithm for computing the gradient of the objective function of the approximated problem. Using the gradient information, we can solve the approximated problem by the projected gradient method. We have also established the local and global convergence of the projected gradient method.

**Limitation and future work**   Our proposed gradient computation algorithm is fixed-parameter tractable and hence it works efficiently for small $n$. For large $n$, however, the exact computation of the gradient is expensive. Development of heuristics for approximately computing the gradient is left for future research. Weakening assumptions in the theoretical contribution is also left for future work. In addition, there is a possibility of another algorithm to solve the approximated problem 5. In this paper, we propose Algorithm 1, which corresponds to forward mode automatic differentiation. On the other hand, in the prior research for bilevel optimization [11], two algorithms were proposed from the perspective of forward mode automatic differentiation and reverse mode automatic differentiation, respectively. Therefore, there is a possibility of another algorithm for problem 5 which corresponds to the reverse mode automatic differentiation, and that is left for future work.

## Acknowledgments and Disclosure of Funding

This work was partially supported by JSPS KAKENHI (JP19K15247, JP17H01699, and JP19H04069).

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
