# Supplementary materials

## A   Proofs

### A.1   Proof of Theorem 2

To prove Theorem 2, we introduce some definitions and a lemma.

**Definition 9** ([15]). Let $\Gamma$ be a point-to-set mapping from a set $\Theta$ into a set $\Xi$, i.e., $\Gamma\colon \Theta \to 2^{\Xi}$.

- $\Gamma$ is said to be *open* at $\bar{\theta} \in \Theta$ if $\{\theta^k\} \subseteq \Theta$, $\theta^k \to \bar{\theta}$, and $\bar{\xi} \in \Gamma(\bar{\theta})$ imply the existence of an integer $m$ and a sequence $\{\xi^k\} \subseteq \Xi$ such that $\xi^k \in \Gamma(\theta^k)$ for $k \geq m$ and $\xi^k \to \bar{\xi}$.

- $\Gamma$ is said to be *closed* at $\bar{\theta} \in \Theta$ if $\{\theta^k\} \subseteq \Theta$, $\theta^k \to \bar{\theta}$, $\xi^k \in \Gamma(\theta^k)$, and $\xi^k \to \bar{\xi}$ imply $\bar{\xi} \in \Gamma(\bar{\theta})$.

- $\Gamma$ is said to be *continuous* at $\bar{\theta} \in \Theta$ if it is both open and closed at $\bar{\theta}$. If $\Gamma$ is continuous at every $\theta \in \Theta$, it is said to be continuous on $\Theta$.

- $\Gamma$ is said to be *uniformly compact* near $\bar{\theta} \in \Theta$ if there is a neighborhood $N$ of $\bar{\theta}$ such that the closure of the set $\bigcup_{\theta \in N} \Omega(\theta)$ is compact.

**Lemma 10** ([15, Corollary 8.1]). *For an objective function $\varphi\colon \mathbb{R}^d \times \Theta \to \mathbb{R}$ and a constraint mapping $\Omega\colon \Theta \to 2^{\mathbb{R}^d}$, consider the following parametric optimization problem with parameter $\theta \in \Theta \subseteq \mathbb{R}^p$:*

$$\min_{\xi} \varphi(\xi, \theta)$$

$$\text{s.t. } \xi \in \Omega(\theta).$$

*For this problem, define the optimal set mapping $\Xi^*\colon \Theta \to 2^{\mathbb{R}^d}$ as*

$$\Xi^*(\theta) := \{\xi \in \Omega(\theta)\colon \varphi(\xi, \theta) = \inf_{\xi' \in \Omega(\theta)} \varphi(\xi', \theta)\}.$$

*Suppose that $\Omega$ is continuous at $\bar{\theta} \in \Theta$, $\varphi$ is continuous on $\Omega(\bar{\theta}) \times \{\bar{\theta}\}$, $\Xi^*$ is nonempty and uniformly compact near $\bar{\theta}$, and $\Xi^*(\bar{\theta})$ is a singleton. Then, $\Xi^*$ is continuous at $\bar{\theta}$.*

*Proof of Theorem 2.* Since $S_1$ is compact from Assumption 1-(i), it follows from the Weierstrass theorem that a sufficient condition for the existence of minimizers is that $F_1$ is continuous. Since $F_1$ is a composition of function $f_1$, which is continuous from Assumption 1-(ii), and functions $x_2^*, \ldots, x_n^*$, we inductively show the continuity of $x_2^*, \ldots, x_n^*$ in what follows.

For any $i = 2, \ldots, n$, we derive the continuity of $x_i^*$ assuming the continuity of $x_{i+1}^*, \ldots, x_n^*$. Since $f_i$ is continuous from Assumption 1-(ii) and so are $x_{i+1}^*, \ldots, x_n^*$ from the induction hypothesis, $F_i$ is continuous. In addition, we define $X_i^*\colon S_1 \times \mathbb{R}^{d_2} \times \cdots \times \mathbb{R}^{d_{i-1}} \to 2^{\mathbb{R}^{d_i}}$ as

$$X_i^*(x_1, \ldots, x_{i-1}) := \left\{x_i \in \mathbb{R}^{d_i}\colon F_i(x_1, \ldots, x_{i-1}, x_i) = \inf_{x_i' \in \mathbb{R}^{d_n}} F_i(x_1, \ldots, x_{i-1}, x_i')\right\}.$$

For any $(\bar{x}_1, \ldots, \bar{x}_{i-1}) \in S_1 \times \mathbb{R}^{d_2} \times \ldots \mathbb{R}^{d_{i-1}}$, $X_i^*$ is nonempty near $(\bar{x}_1, \ldots, \bar{x}_{i-1})$ and $X_i^*(\bar{x}_1, \ldots, \bar{x}_{i-1})$ is a singleton from Assumption 1-(iii); and $X_i^*$ is uniformly bounded near $(\bar{x}_1, \ldots, \bar{x}_{i-1})$ from Assumption 1-(iv). Therefore, from Lemma 10, $X_i^*$ is continuous at $(\bar{x}_1, \ldots, \bar{x}_{i-1})$. From the arbitrariness of $(\bar{x}_1, \ldots, \bar{x}_{i-1})$, $X_i^*$ is continuous on $S_1 \times \mathbb{R}^{d_2} \times \ldots \mathbb{R}^{d_{i-1}}$. Recalling Assumption 1-(iii) again, we have $X_i^*(x_1, \ldots, x_{i-1}) = \{x_i^*(x_1, \ldots, x_{i-1})\}$. The continuity (in the sense of point-to-set mappings) and single-valuedness of $X_i^*$ means the continuity (in the sense of functions) of $x_i^*$. $\qquad\square$

### A.2   Proof of Theorem 4

To prove Theorem 4, we use the following lemma.

**Lemma 11** ([12, Theorem A.1]). *Let $\varphi_T$ and $\varphi$ be continuous[2] functions defined on a compact set $\Omega$. Suppose that $\varphi_T$ converges uniformly to $\varphi$ on $\Omega$ as $T \to \infty$. Then,*

---

[2]In the original statement of [12, Theorem A.1], the lower-semicontinuity is assumed, but in the proof, the continuity is used. [12, Theorem A.1] would be rewriting of [9, Theorem 14]. In the statement of [9, Theorem 14], the continuity is assumed.

*(a)* $\inf_{\xi \in \Omega} \varphi_T(\xi) \to \inf_{\xi \in \Omega} \varphi(\xi)$ *as* $T \to \infty$;

*(b)* $\operatorname{argmin}_{\xi \in \Omega} \varphi_T(\xi) \to \operatorname{argmin}_{\xi \in \Omega} \varphi(\xi)$ *as* $T \to \infty$, *meaning that, for every* $\{\xi_T\}_{T \in \mathbb{N}}$ *such that* $\xi_T \in \operatorname{argmin}_{\xi \in \Omega} \varphi_T(\xi)$, *we have that:*

- $\{\xi_T\}$ *admits a convergent subsequence;*
- *for every subsequence* $\{\xi_{T_k}\}$ *such that* $\xi_{T_k} \to \bar{\xi}$ *as* $k \to \infty$, *we have* $\bar{\xi} \in \operatorname{argmin}_{\xi \in \Omega} \varphi(\xi)$.

*Proof of Theorem 4.* From Assumption 3-(i), there exists $\nu > 0$ such that for every $T \in \mathbb{N}$ and every $x_1 \in S_1$,

$$|\tilde{F}_1(x_1) - F_1(x_1)| = |f_1(x_1, x_2^{(T)}(x_1), \ldots, x_n^{(T)}(x_1)) - f_1(x_1, x_2^*(x_1), \ldots, x_n^*(x_1))|$$

$$\leq \nu \sum_{i=2}^{n} \|x_i^{(T)}(x_1) - x_i^*(x_1)\|.$$

It follows from Assumption 3-(ii) that $\tilde{F}_1$, which depends on $T$, converges to $F_1$ uniformly on $S_1$ as $T \to \infty$. Then, the statement follows from Lemma 11. Note that the continuity of $F_1$ has already shown in Theorem 2. $\quad\square$

## A.3  Proof of Theorem 5

*Proof.* By using the chain rule, we obtain

$$\nabla_{x_1} \tilde{F}_1(x_1) = \nabla_{x_1} f_1 + \sum_{i=2}^{n} \nabla_{x_1} x_i^{(T_i)}(x_1, x_2^{(T_2)}(x_1), \ldots, x_{i-1}^{(T_{i-1})}(x_{k-2}^{(T_{k-2})}(\cdots(x_2^{(T_2)}(x_1))))) \nabla_{x_i} f_1,$$

where we denoted $\nabla_{x_i} f_1 = \nabla_{x_i} f_1(x_1, x_2^{(T_2)}, \ldots, x_n^{(T_n)})$ for $i = 2, \ldots, n$ and $\nabla_{x_1} f_1 = \nabla_{x_1} f_1(x_1, x_2^{(T_2)}, \ldots, x_n^{(T_n)})$. Hereinafter, for $i = 2, \ldots, n$ and $t = 1, \ldots, T_i$, we denote $Y_i^{(t)} = \nabla_{x_1} x_i^{(t)}(x_1, x_2^{(T_2)}(x_1), \ldots, x_{i-1}^{(T_{i-1})}(x_{k-2}^{(T_{k-2})}(\cdots(x_2^{(T_2)}(x_1)))))$, and show $Y_i^{(T_i)} = Z_i$ for all $i = 2, \ldots, n$. In what follows, we denote $\bar{B}_i^{(t)} = \sum_{j=2}^{i-1} Z_j C_{ij}^{(t)} + B_i^{(t)}$.

For $i = 3, \ldots, n$ and $t = 1, \ldots, T_i$, by applying the chain rule to the update formula of $x_i$, i.e., $x_i^{(t)} = \Phi_i^{(t)}(x_1, x_2^{(T_2)}, \ldots, x_{i-1}^{(T_{i-1})}, x_i^{(t-1)})$, we obtain

$$Y_i^{(t)} = Y_i^{(t-1)} \nabla_{x_i} \Phi_i^{(t)}(x_1, x_2^{(T_2)}, \ldots, x_{i-1}^{(T_{i-1})}, x_i^{(t-1)})$$

$$+ \sum_{j=2}^{i-1} Y_j^{(T_j)} \nabla_{x_j} \Phi_i^{(t)}(x_1, x_2^{(T_2)}, \ldots, x_{i-1}^{(T_{i-1})}, x_i^{(t-1)})$$

$$+ \nabla_{x_1} \Phi_i^{(t)}(x_1, x_2^{(T_2)}, \ldots, x_{i-1}^{(T_{i-1})}, x_i^{(t-1)})$$

$$= Y_i^{(t-1)} A_i^{(t)} + \sum_{j=2}^{i-1} Y_j^{(T_j)} C_{ij}^{(t)} + B_i^{(t)}. \tag{9}$$

For $i = 2$ and $t = 1, \ldots, T_2$, note that the arguments of $\Phi_2^{(t)}$ are $x_1$ and $x_2^{(t-1)}$. For $t = 1, \ldots, T_2$, by applying the chain rule, we obtain the following:

$$Y_2^{(t)} = Y_2^{(t-1)} \nabla_{x_2} \Phi_2^{(t)}(x_1, x_2^{(t-1)}) + \nabla_{x_1} \Phi_2^{(t)}(x_1, x_2^{(t-1)})$$

$$= Y_2^{(t-1)} A_2^{(t)} + B_2^{(t)}$$

$$= Y_2^{(t-1)} A_2^{(t)} + \bar{B}_2^{(t)}. \tag{10}$$

Note that, $Y_i^{(0)} = \nabla_{x_1} x_i^{(0)} = O$ for $i = 2, \ldots, n$ since the initial point $x_i^{(0)}$ is independent to the other variables. Using (10) repeatedly, we obtain

$$Y_2^{(T_2)} = Y_2^{(T_2-1)} A_2^{(T_2)} + \bar{B}_2^{(T_2)}$$

$$= (Y_2^{(T_2-2)} A_2^{(T_2-1)} + \bar{B}_2^{(T_2-1)}) A_2^{(T_2)} + \bar{B}_2^{(T_2)}$$

$$= ((Y_2^{(T_2-3)} A_2^{(T_2-2)} + \bar{B}_2^{(T_2-2)}) A_2^{(T_2-1)} + \bar{B}_2^{(T_2-1)}) A_2^{(T_2)} + \bar{B}_2^{(T_2)}$$

$$= \ldots$$

$$= (((\cdots(\bar{B}_2^{(1)} A_2^{(2)} + \bar{B}_2^{(2)}) \cdots) A_2^{(T_2-2)} + \bar{B}_2^{(T_2-2)}) A_2^{(T_2-1)} + \bar{B}_2^{(T_2-1)}) A_2^{(T_2)} + \bar{B}_2^{(T_2)}$$

$$= \sum_{t=1}^{T_2} \bar{B}_2^{(t)} \prod_{s=t+1}^{T_2} A_2^{(s)}$$

$$= Z_2. \tag{11}$$

Next, for $t = 1, \ldots, T_3$, plugging Equation (9) with $i = 3$ into Equation (11), we obtain

$$Y_3^{(t)} = Y_3^{(t-1)} A_3^{(t)} + Z_2 C_{22}^{(t)} + B_3^{(t)} = Y_3^{(t-1)} A_3^{(t)} + \bar{B}_3^{(t)}. \tag{12}$$

Repeatedly using Equation (12) similarly to the derivation of Equation (11), we obtain

$$Y_3^{(T_3)} = \sum_{t=1}^{T_3} \bar{B}_3^{(t)} \prod_{s=t+1}^{T_3} A_3^{(s)} = Z_3.$$

For $i \geq 4$, we can use a similar argument incrementing $i$: When we evaluate $Y_i^{(T_i)}$, since we already have $Y_j^{(T_j)} = Z_j$ for $l = 2, \ldots, k-1$, we can derive the followings:

$$Y_i^{(t)} = Y_i^{(t-1)} A_i^{(t)} + \sum_{j=2}^{i-1} Y_j^{(T_j)} C_{ij}^{(t)} + B_i^{(t)}$$

$$= Y_i^{(t-1)} A_i^{(t)} + \sum_{j=2}^{i-1} Z_j C_{ij}^{(t)} + B_i^{(t)}$$

$$= Y_i^{(t-1)} A_i^{(t)} + \bar{B}_i^{(t)}. \tag{13}$$

Repeatedly using Equation (13) similarly to the derivation of Equation (11), we obtain $Y_i^{(T_i)} = Z_i$ for $i = 2, \ldots, n$. □

### A.4  Proof of Theorem 6

To prove Theorem 6, we introduce the following lemma.

**Lemma 12** (Complexity of Jacobian-vector products [2, 11])**.** *Let $\varphi \colon \mathbb{R}^d \to \mathbb{R}^m$ be a differentiable function and suppose it can be evaluated in time $c(d, m)$ and requires space $s(d, m)$. Denote let $J_\varphi \in \mathbb{R}^{m \times d}$ be the Jacobian matrix of $\varphi$. Then, for any vector $\xi \in \mathbb{R}^d$, the product $J_\varphi \xi$ can be computed within $O(c(d, m))$ time complexity and $O(s(d, m))$ space complexity.*

On the complexity of Algorithm 1, the following lemma holds. Hereinafter, $n$ times nested $O(\cdot)$ is denoted by $O^n(\cdot)$, that is, $O^n(\cdot) = O(O(\cdots O(\cdot) \cdots))$. Note that, for a parameter $n$, $O^n$ is not equivalent to $O$ while $O^c$ is equivalent to $O$ for a constant $c$.

**Lemma 13** (The complexity of Algorithm 1)**.** *Let the time and space complexity for computing $\nabla_{x_i} \tilde{F}_i(x_i)$ be $c_i$ and $s_i$, respectively. We assume that those for evaluating $\Phi_i^{(t)}$ are $O(c_i)$ and $O(s_i)$. In addition, we assume that the time and space complexity of $\nabla_{x_1} f_1$ and $\nabla_{x_i} f_1$ for $i = 1, \ldots, n$ are smaller in the sense of the order than those of for loops in lines 2–9 in Algorithm 1. Then, the time and space complexity of Algorithm 1 can be written as*

$$c_1 = \sum_{i=2}^{n} O(T_i i d_1 \, O(c_i)), \; s_1 = \max_{i=2}^{n} O^2(s_i). \tag{14}$$

*Proof.* For each $i = 2, \ldots, n$ and $t = 1, \ldots, T_i$, the complexity of lines 5–7 of Algorithm 1 can be evaluated as follows:

- From the assumption, the time and space complexity of the 5th line are $O(c_i)$ and $O(s_i)$, respectively.

- The 6th line computes $i - 2$ products of Jacobian matrices and a $d_i \times d_1$ matrices. One product is obtained by computing the product of a Jacobian matrix and a vector $d_1$ times and hence its time and space complexity are $O(d_1 O(c_i))$ and $O(O(s_i))$, respectively. Since the 6th line compute $i - 2$ products, the overall time and space complexity of the 6th line are $O(id_1 O(c_i))$ and $O(O(s_i))$, respectively.

- The 7th line computes the product of a Jacobian matrix and a $d_i \times d_1$ matrix. Its time and space complexity are $O(d_1 O(c_i))$ and $O(O(s_i))$, respectively.

Hence, for each $i = 2, \ldots, n$, the time and space complexity of the single run of the `for` loop in lines 2–7 are $O(T_i i d_1 O(c_i))$, $O(O(s_i))$ respectively. Therefore, we obtain Equation (14) as the overall time and space complexity of Algorithm 1. □

*Proof of Theorem 6.* From the assumption, $c_i$, $s_i$ are the time and space complexity for computing $\nabla_{x_i} \tilde{F}_i(x_i)$. Hence, from Lemma 13, $c_i$ and $s_i$ are written as

$$c_i = \sum_{j=i+1}^{n} O(T_j j d_j O(c_j)), \quad s_i = \max_{j=i+1}^{n} O^2(s_j).$$

From the equation above, $c_i$ ($i = 1, \ldots, n - 1$) can be written by using $c_n$ as follows:

$$
\begin{aligned}
c_{n-1} &= O(T_n n d_{n-1} O(c_n)), \\
c_{n-2} &= O(T_{n-1}(n-1)d_{n-2} O(c_{n-1})) + O(T_n n d_{n-2} O(c_n)) \\
&= O(T_{n-1}(n-1)d_{n-2} O(O(T_n n d_{n-1} O(c_n)))) \\
&= O(T_n T_{n-1} n(n-1) d_{n-1} d_{n-2} O(O(O(c_n)))), \\
&\vdots \\
c_i &= O(T_n T_{n-1} \cdots T_{i+1} n(n-1) \cdots (i+1) d_{n-1} d_{n-2} \cdots d_i O^{2(n-i)-1}(c_n)), \\
&\vdots \\
c_1 &= O(T_n T_{n-1} \cdots T_2 n(n-1) \cdots 2 d_{n-1} d_{n-2} \cdots d_1 O^{2n-3}(c_n)) \\
&= O\left(O^{2n-3}(c_n) n! \prod_{i=1}^{n-1}(T_{i+1} d_i)\right).
\end{aligned}
$$

Similarly, $s_i$ ($i = 1, \ldots, n - 1$) can be written by using $s_n$ as follows:

$$
\begin{aligned}
s_{n-1} &= O^2(s_n), \\
s_{n-2} &= \max_{l=n-1}^{n} O^2(s_l) = \max\{O^2(s_l), O^4(s_l)\} = O^4(s_n), \\
&\vdots \\
s_i &= O^{2(n-i)}(s_n), \\
&\vdots \\
s_1 &= O^{2n-2}(s_n).
\end{aligned}
$$

Note that, since the gradient of the objective function in the lower level is used to calculate the gradient of the objective function in the upper level, the amount of calculation of the gradient at each level increases from the bottom to the top. Therefore, for some $p, q > 1$, it holds that $O^{2n-3}(c_n) = O(p^n c_n)$ and $O^{2n-2}(s_n) = O(q^n s_n)$. Hence, we obtain Equation (7) as the overall time and space complexity for computing $\nabla_{x_1} \tilde{F}_1(x_1)$ recursively calling Algorithm 1. □

### A.5 Proof of Theorem 7

*Proof.* In this proof, we inductively show the Lipschitz continuity of $\nabla_{x_i} \tilde{F}_i$ for all $i = 1, \ldots, n$ by using the following notations: When a function $\varphi$ is Lipschitz continuous on $\Omega$, its Lipschitz constant

is denoted by $L_\varphi$, i.e., $\|\varphi(\hat\xi) - \varphi(\check\xi)\| \le L_\varphi \|\hat\xi - \check\xi\|$ for any $\hat\xi, \check\xi \in \Omega$. When $\varphi$ is bounded on $\Omega$, its bound is denoted by $M_\varphi$, i.e., $\|\varphi(\xi)\| \le M_\varphi$ for any $\xi \in \Omega$.

As the base case, we consider the case of $i = n$. Since $\tilde F_n = f_n$, the Lipschitz continuity of $\nabla_{x_n} \tilde F_n$ follows from the assumption of the Lipschitz continuity of $\nabla_{x_i} f_n$.

In what follows, as the induction step, we derive the Lipschitz continuity of $\nabla_{x_i} \tilde F_i$ by assuming the Lipschitz continuity of $\nabla_{x_j} \tilde F_j$ for $j = i+1, \ldots, n$. To do so, for any $(\hat x_1, \ldots, \hat x_i)$ and $(\check x_1, \ldots, \check x_i)$ in $S_1 \times \mathbb{R}^{d_2} \times \cdots \times \mathbb{R}^{d_i}$, we evaluate $\|\nabla_{x_i} \tilde F_i(\hat x_1, \ldots, \hat x_i) - \nabla_{x_i} \tilde F_i(\check x_1, \ldots, \check x_i)\|$. In what follows, constants related to $(\hat x_1, \ldots, \hat x_i)$ and $(\check x_1, \ldots, \check x_i)$ are denoted by ones with hat and check sign, e.g., $\hat Z_j$ and $\check Z_j$. By applying Theorem 5 to $\tilde F_i$, we have

$$
\begin{aligned}
&\|\nabla_{x_i} \tilde F_i(\hat x_1, \ldots, \hat x_i) - \nabla_{x_i} \tilde F_i(\check x_1, \ldots, \check x_i)\| \\
&\le \|\nabla_{x_i} f_i(\hat x_1, \ldots, \hat x_i, \hat x_{i+1}^{(T_{i+1})}, \ldots, \hat x_n^{(T_n)}) - \nabla_{x_i} f_i(\check x_1, \ldots, \check x_i, \check x_{i+1}^{(T_{i+1})}, \ldots, \check x_n^{(T_n)})\| \\
&\quad + \sum_{j=2}^n \|\hat Z_j \nabla_{x_j} f_i(\hat x_1, \ldots, \hat x_i, \hat x_{i+1}^{(T_{i+1})}, \ldots, \hat x_n^{(T_n)}) - \check Z_j \nabla_{x_j} f_i(\check x_1, \ldots, \check x_i, \check x_{i+1}^{(T_{i+1})}, \ldots, \check x_n^{(T_n)})\|
\end{aligned}
\tag{15}
$$

First, we evaluate the first term of the right-hand-side of Equation (15). From the assumption of the Lipschitz continuity of $\nabla_{x_i} f_i$, we have

$$
\begin{aligned}
&\|\nabla_{x_i} f_i(\hat x_1, \ldots, \hat x_i, \hat x_{i+1}^{(T_{i+1})}, \ldots, \hat x_n^{(T_n)}) - \nabla_{x_i} f_i(\check x_1, \ldots, \check x_i, \check x_{i+1}^{(T_{i+1})}, \ldots, \check x_n^{(T_n)})\| \\
&\le L_{\nabla_{x_i} f_i} \left( \|(\hat x_1, \ldots, \hat x_i) - (\check x_1, \ldots, \check x_i)\| + \sum_{j=i+1}^n \|\hat x_j^{(T_j)} - \check x_j^{(T_j)}\| \right).
\end{aligned}
$$

For any $j = i+1, \ldots, n$ and $t = 1, \ldots, T_j$, recalling the definition of $\hat x_j^{(t)}$ and $\check x_j^{(t)}$ and invoking the Lipschitz continuity of $\nabla_{x_j} \tilde F_j$, we have

$$
\begin{aligned}
&\|\hat x_j^{(t)} - \check x_j^{(t)}\| \\
&= \|\Phi_j^{(t)}(\hat x_1, \ldots, \hat x_i, \hat x_{i+1}^{(T_{i+1})}, \ldots, \hat x_{j-1}^{(T_{j-1})}, \hat x_j^{(t-1)}) - \Phi_j^{(t)}(\check x_1, \ldots, \check x_i, \check x_{i+1}^{(T_{i+1})}, \ldots, \check x_{j-1}^{(T_{j-1})}, \check x_j^{(t-1)})\| \\
&\le \|\hat x_j^{(t-1)} - \check x_j^{(t-1)}\| + \alpha_j^{(t-1)} \left\| \begin{aligned} &\nabla_{x_j} \tilde F_j(\hat x_1, \ldots, \hat x_i, \hat x_{i+1}^{(T_{i+1})}, \ldots, \hat x_{j-1}^{(T_{j-1})}, \hat x_j^{(t-1)}) \\ &- \nabla_{x_j} \tilde F_j(\check x_1, \ldots, \check x_i, \check x_{i+1}^{(T_{i+1})}, \ldots, \check x_{j-1}^{(T_{j-1})}, \check x_j^{(t-1)}) \end{aligned} \right\| \\
&\le \|\hat x_j^{(t-1)} - \check x_j^{(t-1)}\| + \alpha_j^{(t-1)} L_{\nabla_{x_j} \tilde F_j} \left( \begin{aligned} &\|(\hat x_1, \ldots, \hat x_i) - (\check x_1, \ldots, \check x_i)\| \\ &+ \sum_{k=i+1}^{j-1} \|\hat x_k^{(T_k)} - \check x_k^{(T_k)}\| + \|\hat x_j^{(t-1)} - \check x_j^{(t-1)}\| \end{aligned} \right).
\end{aligned}
$$

In addition, for any $j = i+1, \ldots, n$, we have

$$
\begin{aligned}
&\|\hat x_j^{(1)} - \check x_j^{(1)}\| \\
&= \|\Phi_j^{(1)}(\hat x_1, \ldots, \hat x_i, \hat x_{i+1}^{(T_{i+1})}, \ldots, \hat x_{j-1}^{(T_{j-1})}, x_j^{(0)}) - \Phi_i^{(1)}(\check x_1, \ldots, \check x_i, \check x_{i+1}^{(T_{i+1})}, \ldots, \check x_{j-1}^{(T_{j-1})}, \check x_j^{(0)})\| \\
&\le \underbrace{\|x_j^{(0)} - \check x_j^{(0)}\|}_{=0} + \alpha_j^{(0)} L_{\nabla_{x_i} \tilde F_i} \left( \begin{aligned} &\|(\hat x_1, \ldots, \hat x_i) - (\check x_1, \ldots, \check x_i)\| \\ &+ \sum_{k=i+1}^{j-1} \|\hat x_k^{(T_k)} - \check x_k^{(T_k)}\| + \underbrace{\|\hat x_j^{(0)} - \check x_j^{(0)}\|}_{=0} \end{aligned} \right).
\end{aligned}
$$

Using these inequalities repeatedly, we can derive

$$
\begin{aligned}
&\|\nabla_{x_i} f_i(\hat x_1, \ldots, \hat x_i, \hat x_{i+1}^{(T_{i+1})}, \ldots, \hat x_n^{(T_n)}) - \nabla_{x_i} f_i(\check x_1, \ldots, \check x_i, \check x_{i+1}^{(T_{i+1})}, \ldots, \check x_n^{(T_n)})\| \\
&\le N_1 \|(\hat x_1, \ldots, \hat x_i) - (\check x_1, \ldots, \check x_i)\|
\end{aligned}
$$

for some positive constant $N_1$ depending only on $\{\alpha_j^{(t-1)}\}$, $L_{\nabla_{x_i} f_i}$ and $L_{\nabla_{x_i} \tilde{F}_j}$.

Next, we evaluate each summand in the second term of the right-hand-side of Equation (15). To do so, we notice that $\hat{Z}_j$ ($\check{Z}_j$, respectively) is the sum of products of $\hat{A}_j^{(t)}$'s, $\hat{B}_j^{(t)}$'s, and $\hat{C}_{jk}^{(t)}$'s ($\check{A}_j^{(t)}$'s, $\check{B}_j^{(t)}$'s, and $\check{C}_{jk}^{(t)}$'s, respectively). Formally, we can decompose $\hat{Z}_j = \sum_{p=1}^{P} \prod_{q=1}^{Q_p} \hat{X}_{jpq}$, where $\hat{X}_{jpq} \in \bigcup_{t=1}^{T_j}(\{\hat{A}_j^{(t)}, \hat{B}_j^{(t)}\} \cup \bigcup_{k=1}^{j-1}\{\hat{C}_{jk}^{(t)}\})$. Similarly, by appropriately defining $\check{X}_{jpq}$, we can rewrite $\check{Z}_j = \sum_{p=1}^{P} \prod_{q=1}^{Q_p} \check{X}_{jpq}$, where $\check{X}_{jpq} \in \bigcup_{t=1}^{T_j}(\{\check{A}_j^{(t)}, \check{B}_j^{(t)}\} \cup \bigcup_{k=1}^{j-1}\{\check{C}_{jk}^{(t)}\})$ and $\check{X}_{jpq} = \check{A}_j^{(t)}, \check{B}_j^{(t)}, \check{C}_{jk}^{(t)}$ if $\hat{X}_{jpq} = \hat{A}_j^{(t)}, \hat{B}_j^{(t)}, \hat{C}_{jk}^{(t)}$, respectively. Using this notation, we have

$$\|\hat{Z}_j \nabla_{x_j} f_i(\hat{x}_1, \ldots, \hat{x}_i, \hat{x}_{i+1}^{(T_{i+1})}, \ldots, \hat{x}_n^{(T_n)}) - \check{Z}_j \nabla_{x_j} f_i(\check{x}_1, \ldots, \check{x}_i, \check{x}_{i+1}^{(T_{i+1})}, \ldots, \check{x}_n^{(T_n)})\|$$

$$\leq \sum_{p=1}^{P} \left\| \prod_{q=1}^{Q_p} \hat{X}_{jpq} \nabla_{x_j} f_i(\hat{x}_1, \ldots, \hat{x}_i, \hat{x}_{i+1}^{(T_{i+1})}, \ldots, \hat{x}_n^{(T_n)}) - \prod_{q=1}^{Q_p} \check{X}_{jpq} \nabla_{x_j} f_i(\check{x}_1, \ldots, \check{x}_i, \check{x}_{i+1}^{(T_{i+1})}, \ldots, \check{x}_n^{(T_n)}) \right\|$$

$$\leq \sum_{p=1}^{P} \left( \sum_{r=1}^{Q_p} \left\| \prod_{q=1}^{r-1} \check{X}_{jpq} \prod_{q=r}^{Q_p} \hat{X}_{jpq} \nabla_{x_j} f_i(\hat{x}_1, \ldots, \hat{x}_i, \hat{x}_{i+1}^{(T_{i+1})}, \ldots, \hat{x}_n^{(T_n)}) \right. \right.$$
$$\left. \left. - \prod_{q=1}^{r} \check{X}_{jpq} \prod_{q=r+1}^{Q_p} \hat{X}_{jpq} \nabla_{x_j} f_i(\hat{x}_1, \ldots, \hat{x}_i, \hat{x}_{i+1}^{(T_{i+1})}, \ldots, \hat{x}_n^{(T_n)}) \right\| \right.$$
$$\left. + \left\| \prod_{q=1}^{Q_p} \check{X}_{jpq} \nabla_{x_j} f_i(\hat{x}_1, \ldots, \hat{x}_i, \hat{x}_{i+1}^{(T_{i+1})}, \ldots, \hat{x}_n^{(T_n)}) \right. \right.$$
$$\left. \left. - \prod_{q=1}^{Q_p} \check{X}_{jpq} \nabla_{x_j} f_i(\check{x}_1, \ldots, \check{x}_i, \check{x}_{i+1}^{(T_{i+1})}, \ldots, \check{x}_n^{(T_n)}) \right\| \right)$$

$$\leq \sum_{p=1}^{P} \left( \sum_{r=1}^{Q_p} \|\hat{X}_{jpr} - \check{X}_{jpr}\| \|\nabla_{x_j} f_i(\hat{x}_1, \ldots, \hat{x}_i, \hat{x}_{i+1}^{(T_{i+1})}, \ldots, \hat{x}_n^{(T_n)})\| \prod_{q=1}^{r-1} \|\check{X}_{jpq}\| \prod_{q=r+1}^{Q_p} \|\hat{X}_{jpq}\| \right.$$
$$\left. + \prod_{q=1}^{Q_p} \|\check{X}_{jpq}\| \left\| \nabla_{x_j} f_i(\hat{x}_1, \ldots, \hat{x}_i, \hat{x}_{i+1}^{(T_{i+1})}, \ldots, \hat{x}_n^{(T_n)}) - \nabla_{x_j} f_i(\check{x}_1, \ldots, \check{x}_i, \check{x}_{i+1}^{(T_{i+1})}, \ldots, \check{x}_n^{(T_n)}) \right\| \right)$$

$$\leq \underbrace{\sum_{p=1}^{P} \left( \sum_{r=1}^{Q_p} L_{X_{jpr}} M_{\nabla_{x_j} f_i} \prod_{q \neq r} M_{X_{jpq}} + L_{\nabla_{x_j} f_i} \prod_{q=1}^{Q_p} M_{X_{jpq}} \right)}_{= N_2} \|(\hat{x}_1, \ldots, \hat{x}_i) - (\check{x}_1, \ldots, \check{x}_i)\|,$$

where, in the last inequality, we used the Lipschitz continuity and the boundedness of $\nabla_{x_i} \Phi_i^{(t)}$, $\nabla_{x_1} \Phi_i^{(t)}$, and $\nabla_{x_j} \Phi_i^{(t)}$. Therefore, we obtain

$$\|\nabla_{x_i} \tilde{F}_i(\hat{x}_1, \ldots, \hat{x}_i) - \nabla_{x_i} \tilde{F}_i(\check{x}_1, \ldots, \check{x}_i)\| \leq \left( N_1 + \sum_{j=2}^{n} N_{2j} \right) \|(\hat{x}_1, \ldots, \hat{x}_i) - (\check{x}_1, \ldots, \check{x}_i)\|.$$

This means the Lipschitz continuity of $\nabla_{x_i} \tilde{F}_i$, and hence, that of $\nabla_{x_1} \tilde{F}_1$ is obtained by induction. $\quad\square$

# B An example of applying Algorithm 1

We explain an example of approximated problems to be solved by Algorithm 1 (i.e., problem 5 with $n = 3$) by assuming a simple setting, where we apply the steepest descent method for the lower-level

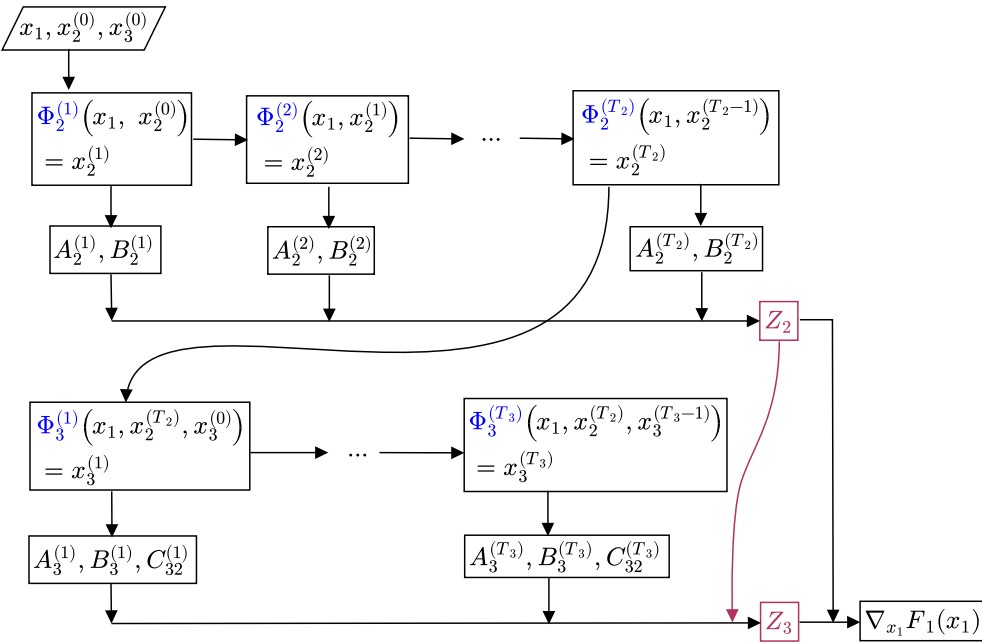

Figure 5: Procedure when applying proposed algorithm to trilevel optimization problems.

problems with the same iteration number $T$ and step size $\alpha$ for all levels as follows:

$$
\begin{aligned}
\min_{x_1 \in S_1, \{x_2^{(t)}\}, \{x_3^{(t)}\}} \quad & f_1(x_1, x_2^{(T)}, x_3^{(T)}) \\
\text{s.t.} \quad & x_2^{(t)} = x_2^{(t-1)} - \alpha \nabla_{x_2} \tilde{F}_2(x_1, x_2^{(t-1)}), && (t = 1, \dots, T), \\
& x_3^{(t)} = x_3^{(t-1)} - \alpha \nabla_{x_3} \tilde{F}_3(x_1, x_2^{(T)}, x_3^{(t-1)}) && (t = 1, \dots, T).
\end{aligned}
\tag{16}
$$

Here, $\nabla_{x_2} \tilde{F}_2(x_1, x_2)$ is the gradient of the objective function of $\min_{x_2 \in \mathbb{R}^{d_2}} \tilde{F}_2(x_1, x_2)$, which is equivalent to the bilevel optimization problem as follows:

$$
\begin{aligned}
\min_{x_2^{(t)}, \{x_3^{(t')}\}} \quad & f_2(x_1, x_2^{(t)}, x_3^{(T)}) \\
\text{s.t.} \quad & x_3^{(t')} = x_3^{(t'-1)} - \alpha \nabla_{x_3} \tilde{F}_3(x_1, x_2^{(t)}, x_3^{(t'-1)}) \quad (t' = 1, \dots, T).
\end{aligned}
$$

In addition, since the third level problem is the lowest level problem, $\tilde{F}_3 = f_3$ holds. We replace $x_2^{(T)}$ in the objective function of (16) by $x_2^{(T-1)} - \alpha \nabla_{x_2} \tilde{F}_2(x_1, x_2^{(T-1)})$ and then replace recursively $x_2^{(t)}$ using $\{x_2^{(t)}\}_{t=0}^{T-1}$. By applying the same procedure for $x_3^{(T)}$, we can reformulate (16) to $\min_{x_1 \in S_1} \tilde{F}_1(x_1)$. Theorem 4 proves that this problem converges to the trilevel optimization problem as $T \to \infty$.

The gradient of $\tilde{F}_1(x_1)$ can be calculated by applying Algorithm 1 to this problem. The explicit formula of $\nabla_{x_1} \tilde{F}_1(x_1)$ is given in Theorem 5 and hence we can compute it by using Algorithm 1. To compute $\nabla_{x_1} \tilde{F}_1(x_1)$ with Algorithm 1, the computation of $\nabla_{x_2} \tilde{F}_2(x_1, x_2)$ is also required. Its explicit formula is also given in Theorem 5 and hence we can compute it by using Algorithm 1. To compute $\nabla_{x_2} \tilde{F}_2(x_1, x_2)$ with Algorithm 1, the computation of $\nabla_{x_3} \tilde{F}_3(x_1, x_2, x_3)$ is also required. This computation is easy (recall $\tilde{F}_3 = f_3$). With these results, we can apply a gradient-based method (e.g., the projected gradient method) to minimize $\tilde{F}_1(x_1)$ using $\nabla \tilde{F}_1$. This procedure of gradient computation is depicted in Figure 5.

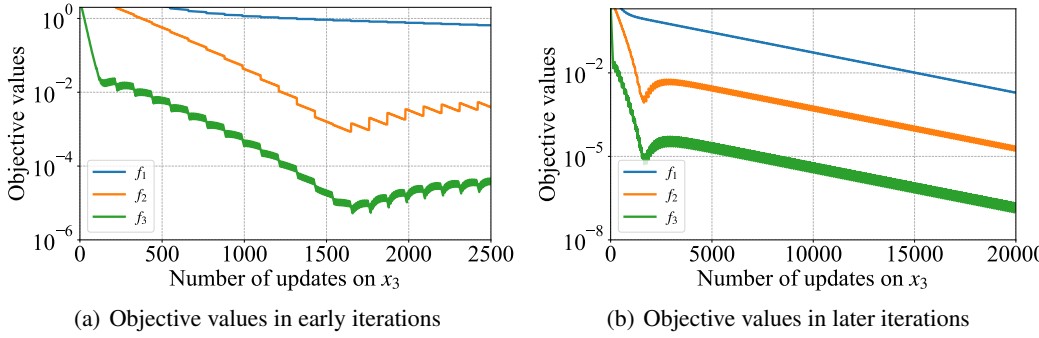

(a) Objective values in early iterations

(b) Objective values in later iterations

Figure 6: Performance of Algorithm 1 for Problem (17) ($T_2 = 10$, $T_3 = 10$).

## C Complete numerical results

To validate the effectiveness of our proposed method, we conducted numerical experiments on an artificial problem and a hyperparameter optimization problem arising from real data. In our numerical experiments, we implemented all codes with Python 3.9.2 and JAX 0.2.10 for automatic differentiation and executed them on a computer with 12 cores of Intel Core i7-7800X CPU 3.50 GHz, 64 GB RAM, Ubuntu OS 20.04.2 LTS.

### C.1 Convergence to optimal solution

We solved the following trilevel optimization problem by using Algorithm 1 to evaluate the performance:

$$\min_{x_1 \in \mathbb{R}^2} f_1(x_1, x_2^*, x_3^*) = \|x_3^* - x_1\|_2^2 + \|x_1\|_2^2 \text{ s.t.}$$

$$x_2^* \in \operatorname*{argmin}_{x_2 \in \mathbb{R}^2} f_2(x_1, x_2, x_3^*) = \|x_2 - x_1\|_2^2 \text{ s.t.} \tag{17}$$

$$x_3^* \in \operatorname*{argmin}_{x_3 \in \mathbb{R}^2} f_3(x_1, x_2, x_3) = \|x_3 - x_2\|_2^2.$$

Clearly, the optimal solution for this problem is $x_1 = x_2 = x_3 = (0, 0)^\top$.

We solved (8) with fixed constant step size and initialization but different $(T_2, T_3)$. For the iterative method in Algorithm 1, we employed the steepest descent method at all levels. We show the transition of the value of the objective functions in Figures 6, 7, 8, 9, and 10 and the trajectories of each decision variables in Figure 11. We can confirm that the objective values $f_1$, $f_2$, and $f_3$ converge to the optimal value 0 at all levels and the gradient method outputs the optimal solution $(0, 0)^\top$ even if we use small iteration numbers $T_2 = 1$ and $T_3 = 1$ for the approximation problem (5). In Figures 6, 7, 8, and 9, there is oscillation of the objective values of lower levels, as some decision variable moves away and closer to other decision variables. In Figures 6, 7, and 9, there is a temporary trade-off in the objective values between the levels.

### C.2 Comparison between Algorithm 1 and an existing algorithm

We also conducted an experiments to compare the performance of Algorithm 1 with that of an existing method [23], which is based on evolutional strategy. In this experiment, we used the same settings in Section 5.1.

First, we applied the evolutional strategic algorithm [23, Table 1 and Table 2] to the problem 8. To clarify its difference between Algorithm 1, we picked the same initial solution as that of Section 5.1. In [23, EvolutionaryStrategy in Table 2], the number of the generations of solution candidates and the number of iterations of perturbations were both set to be 10, the algorithm parameter $\delta$ was set to be $10^{-2}$ and $10^{-4}$, and the solution candidates were randomly chosen from the standard normal distribution. We conducted the experiment 100 times for each $\delta$, and we show the transition of the average of the objective functions and the trajectories of each decision variables in Figures 12 and 13.

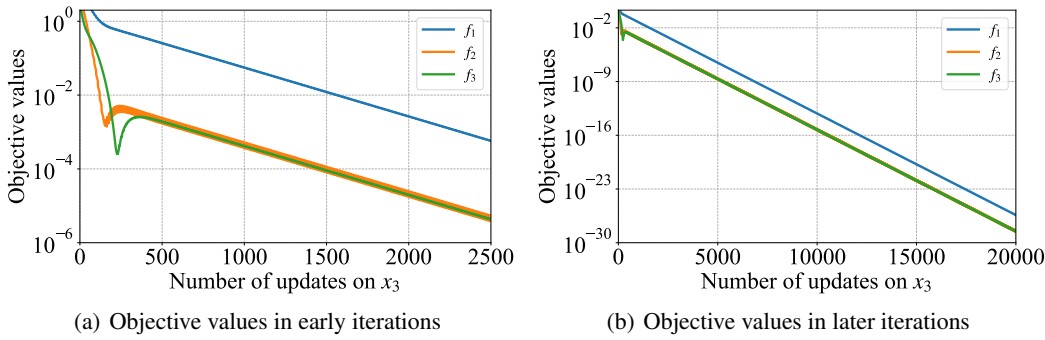

(a) Objective values in early iterations

(b) Objective values in later iterations

Figure 7: Performance of Algorithm 1 for Problem (17) ($T_2 = 10$, $T_3 = 1$).

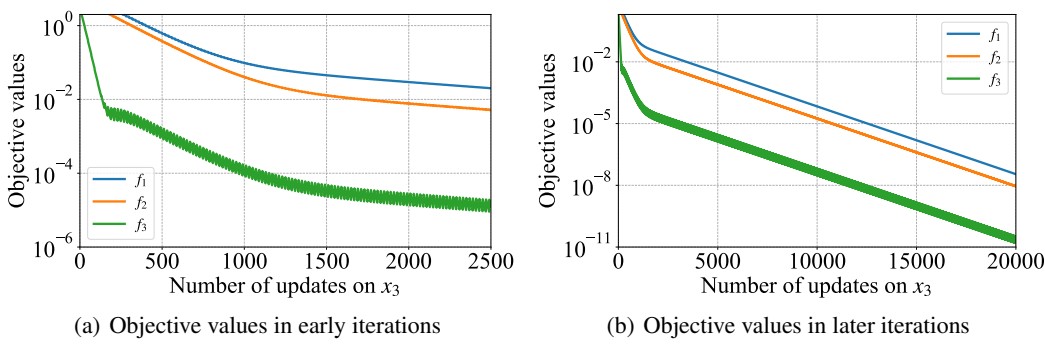

(a) Objective values in early iterations

(b) Objective values in later iterations

Figure 8: Performance of Algorithm 1 for Problem (17) ($T_2 = 1$, $T_3 = 10$).

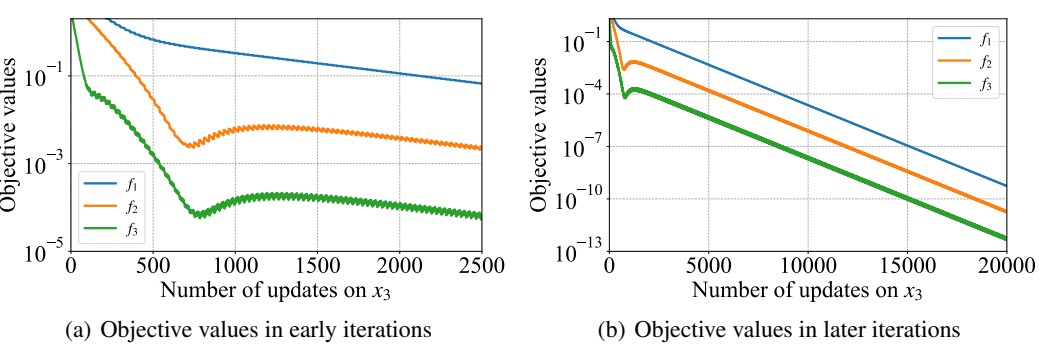

(a) Objective values in early iterations

(b) Objective values in later iterations

Figure 9: Performance of Algorithm 1 for Problem (17) ($T_2 = 5$, $T_3 = 5$).

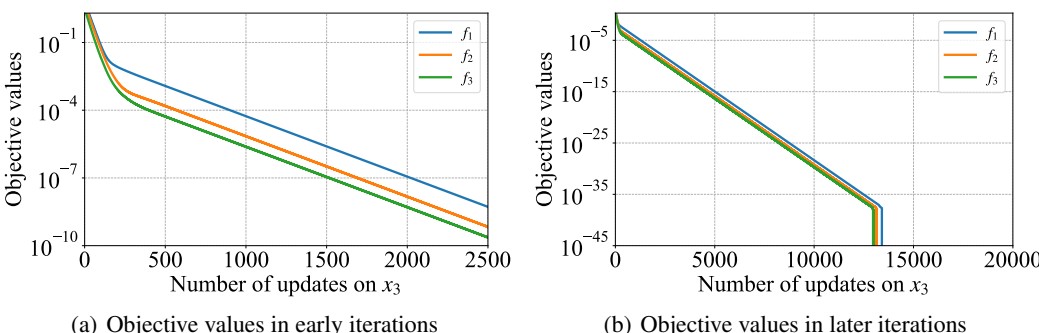

(a) Objective values in early iterations

(b) Objective values in later iterations

Figure 10: Performance of Algorithm 1 for Problem (17) ($T_2 = 1$, $T_3 = 1$).

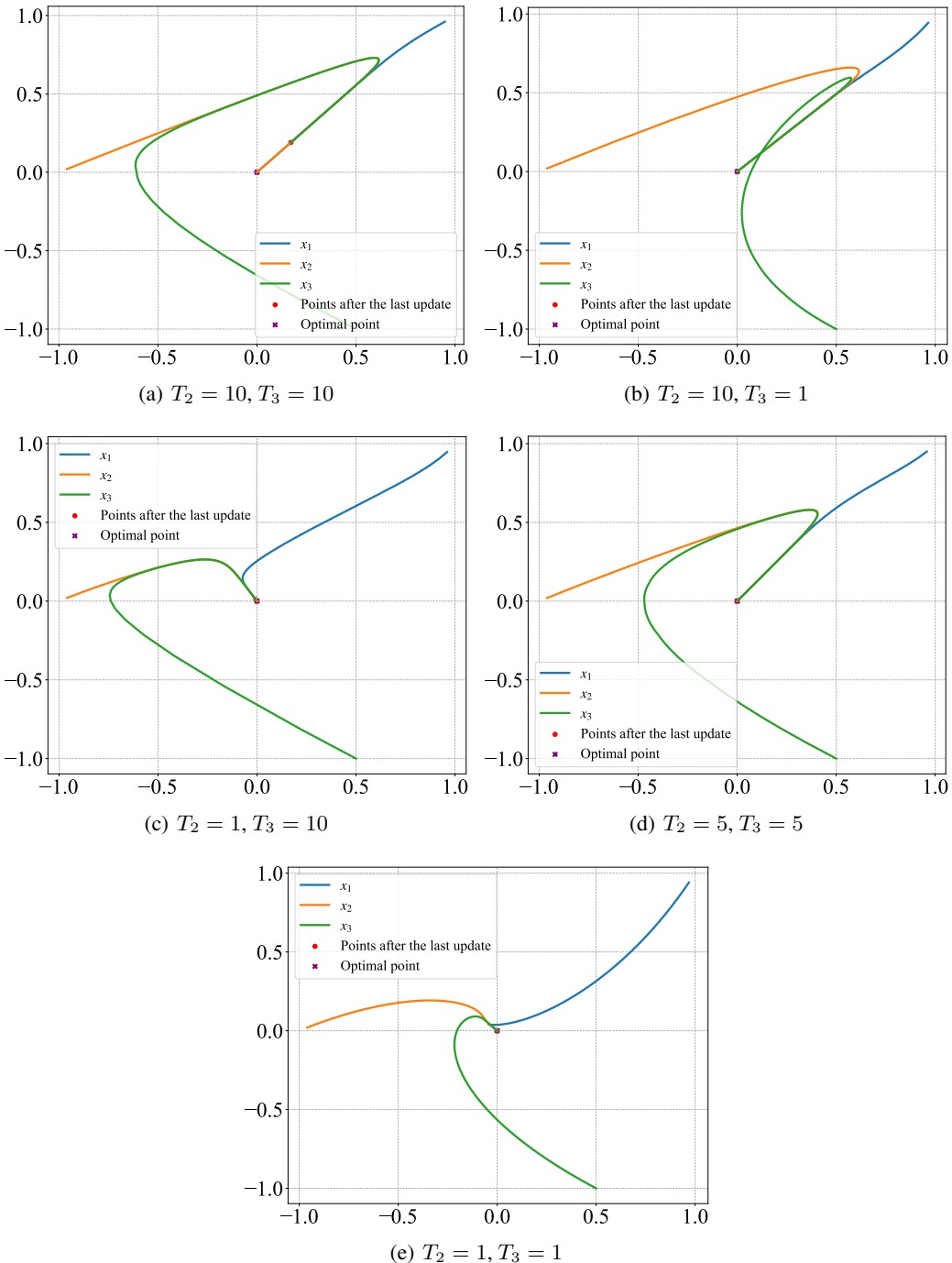

Figure 11: Trajectories of each variables.

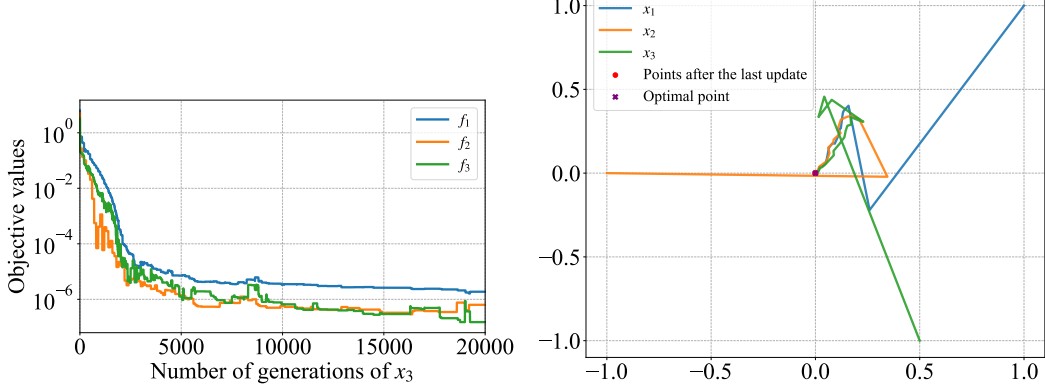

(a) Transition of objective values (average of 100 times)    (b) Trajectory of each variables (one of 100 times)

Figure 12: Performance of the evolutional strategic algorithm [23] for Problem (17) ($\delta = 10^{-2}$).

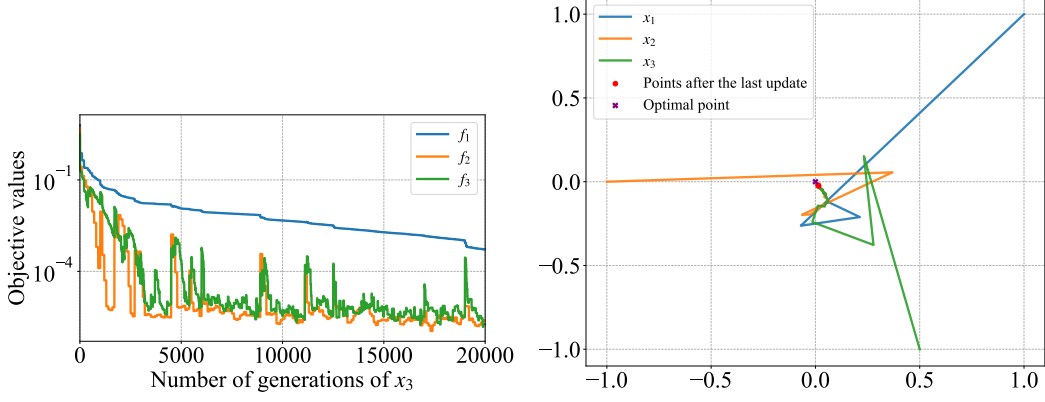

(a) Transition of objective values (average of 100 times)    (b) Trajectory of each variables (one of 100 times)

Figure 13: Performance of the evolutional strategic algorithm [23] for Problem (17) ($\delta = 10^{-4}$).

Next, we compared the performance of the existing method with that of Algorithm 1. We show the transition of the objective values when using each algorithm in Figure 14 by overlaying Figures 6(b), 10(b), 12(a), and 13(a). In later iterations, we can confirm that the convergence of Algorithm 1 with $(T_2, T_3) = (1, 1)$ is faster than the evolutional strategic algorithm.

### C.3 Application to hyperparameter optimization

Next, we conducted experiments on real data to usefulness of our method for a machine learning problem. We consider the problem of how to achieve a model robust to noise in input data and formulate this problem by a trilevel model by a model learner and an attacker. The model learner decides the hyperparameter $\lambda$ to minimize the validation error, while the attacker tries to poison training data so as to make the model less accurate. This model is formulated as follows:

$$\min_{\lambda} \frac{1}{m} \|y_{\text{valid}} - f(X_{\text{valid}}; \theta)\|_2^2 \text{ s.t.}$$

$$P \in \operatorname*{argmax}_{P'} \frac{1}{n} \|y_{\text{train}} - f(X_{\text{train}} + P'; \theta)\|_2^2 - \frac{c}{nd} \|P'\|_2^2 \text{ s.t.}$$

$$\theta \in \operatorname*{argmin}_{\theta'} \frac{1}{n} \|y_{\text{train}} - f(X_{\text{train}} + P'; \theta')\|_2^2 + \exp(\lambda) \frac{\|\theta'\|_{1*}}{d},$$

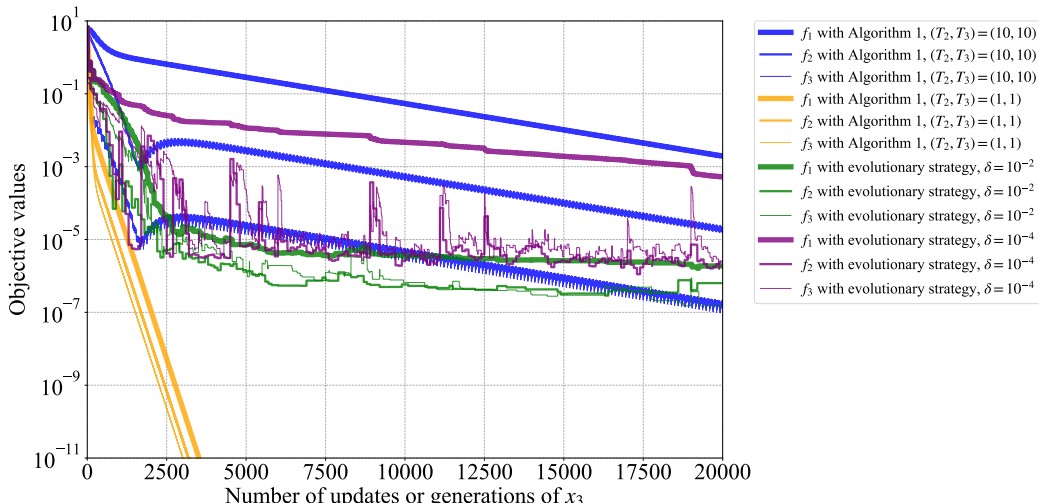

Figure 14: Comparison between the evolutional strategy [23] and Algorithm 1 for Problem (17). Transition of the objective values is plotted. Note that the objective values with the evolutionary strategy are the average of 100 times.

where $f$ denotes the output of a three-layer perceptron which has 3 hidden units, $\theta$ denotes the parameter of the model $f$, $d$ denotes the dimension of $\theta$, $n$ denotes the number of the training data $X_\text{train}$, $m$ denotes the number of the validation data $X_\text{val}$, $c$ denotes the penalty for the noise $P$, and $\|\cdot\|_{1*}$ is a smoothed $\ell_1$-norm, which is a differentiable approximation of the $\ell_1$-norm. We used the hyperbolic tangent function as the activation function for the hidden layer of the multilayer perceptron. Here, we use $\exp(\lambda)$ to express nonnegative penalty parameter instead of the constraint $\lambda \geq 0$.

To validate the effectiveness of our proposed method, we compared the results by the trilevel model with those of the following bilevel model:

$$\min_\lambda \frac{1}{m}\|y_\text{valid} - f(X_\text{valid}; \theta)\|_2^2 \text{ s.t. } \theta \in \operatorname*{argmin}_{\theta'} \frac{1}{n}\|y_\text{train} - f(X_\text{train}; \theta')\|_2^2 + \exp(\lambda)\frac{\|\theta'\|_{1*}}{d}.$$

This model is equivalent to the trilevel model without the attacker's level.

We used Algorithm 1 to compute the gradient of the objective function in the trilevel and bilevel models with real datasets. For the iterative method in Algorithm 1, we employed the steepest descent method at all levels. We set $T_2 = 30$ and $T_3 = 3$ for the trilevel model and $T_2 = 30$ for the bilevel model. In each dataset, we used the same initialization and step sizes in the updates of $\lambda$ and $\theta$ in trilevel and bilevel models. We compared these methods on the regression tasks with the following datasets: the diabetes dataset [10], the (red and white) wine quality datasets [8], the Boston dataset [14]. For each dataset, we standardized each feature and the objective variable; and randomly chose 40 rows as training data $(X_\text{train}, y_\text{train})$ other chose 100 rows as validation data $(X_\text{valid}, y_\text{valid})$, and used the rest of the rows as test data.

We show the transition of the MSE of test data with Gaussian noise in Figures 15, 16, 17, and 18. The solid line and colored belt respectively indicate the mean and the standard deviation over 500 times of generation of Gaussian noise. The dashed line indicates the mean squared error (MSE) without noise as a baseline. In the results of the diabetes dataset (Figure 15), the trilevel model provided a more robust parameter than the bilevel model, because the MSE rises less with the large standard deviation of noise on test data.

Next, we compared the quality of the resulting model parameters by the trilevel and bilevel models. We set an early-stopping condition on learning parameters: after 1000 times of updates on the model parameter, if one time of update on hyperparameter $\lambda$ did not improve test error by $\epsilon$, terminate the iteration and return the parameters at that time. By using the early-stopped parameters obtained by this stopping condition, we show the relationship of test error and standard deviation of the noise on the test data in Figure 19 and Table 2. For the diabetes dataset, the growth of MSE of the trilevel model was slower than that of the bilevel model. For wine quality and Boston house-prices datasets,

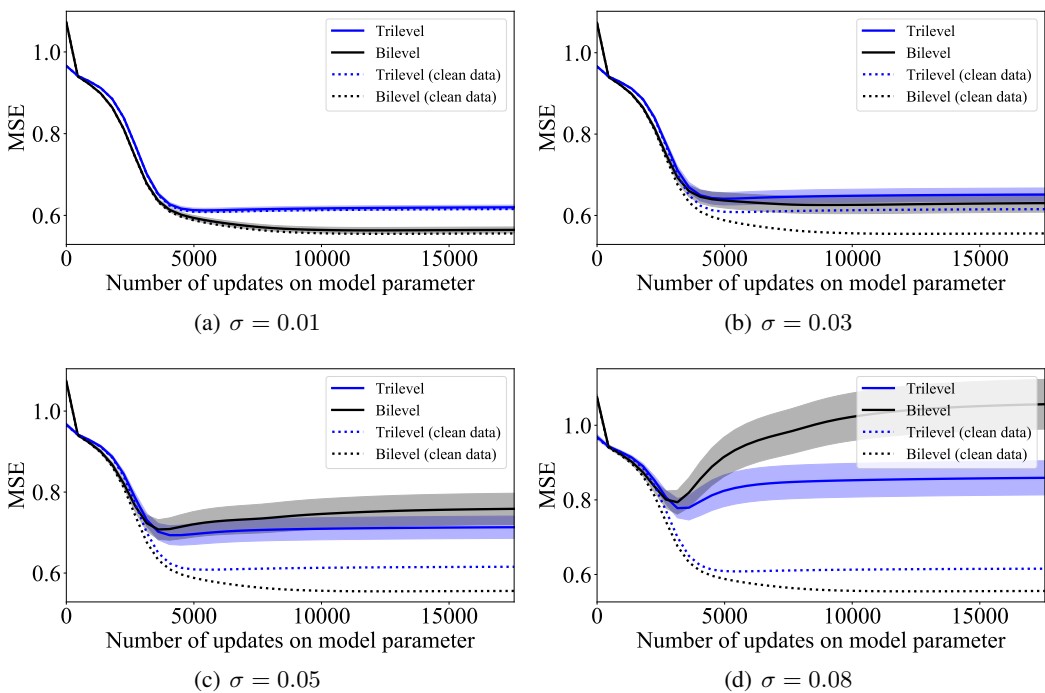

Figure 15: MSE of test data with Gaussian noise with a standard deviation of $\sigma$ on diabetes dataset.

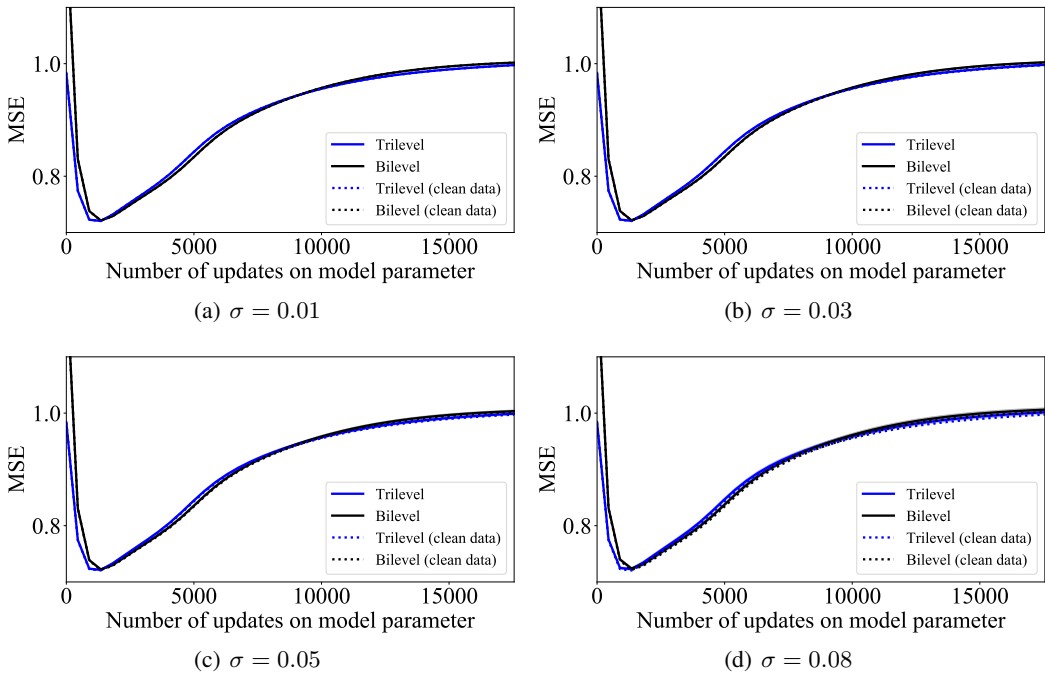

Figure 16: MSE of test data with Gaussian noise with a standard deviation of $\sigma$ on red wine quality dataset.

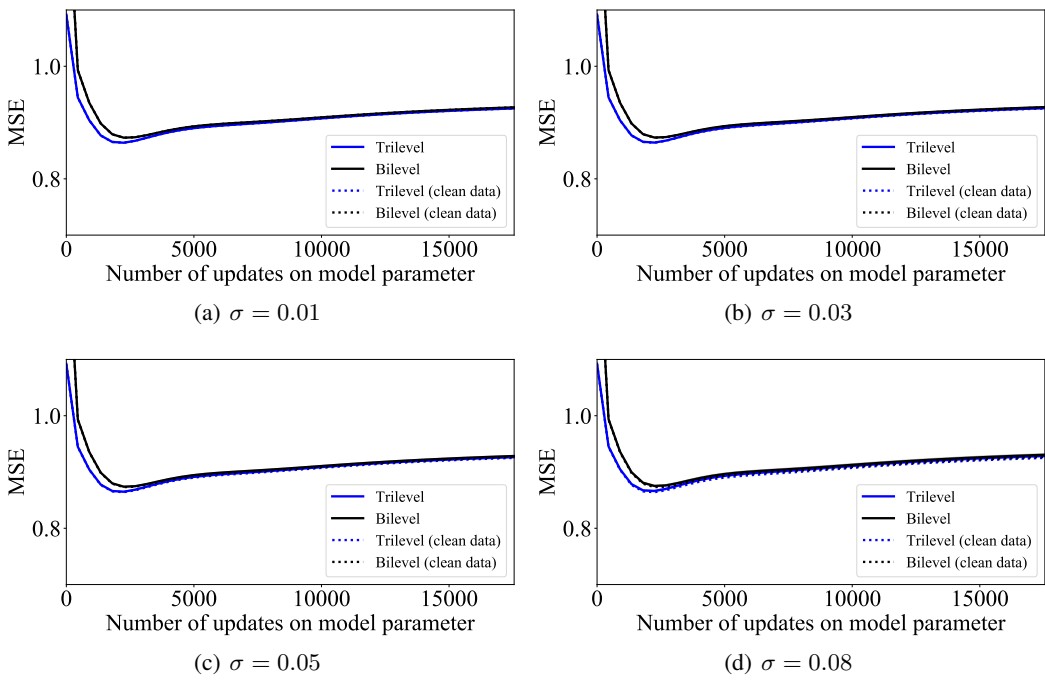

Figure 17: MSE of test data with Gaussian noise with a standard deviation of $\sigma$ on white wine quality dataset.

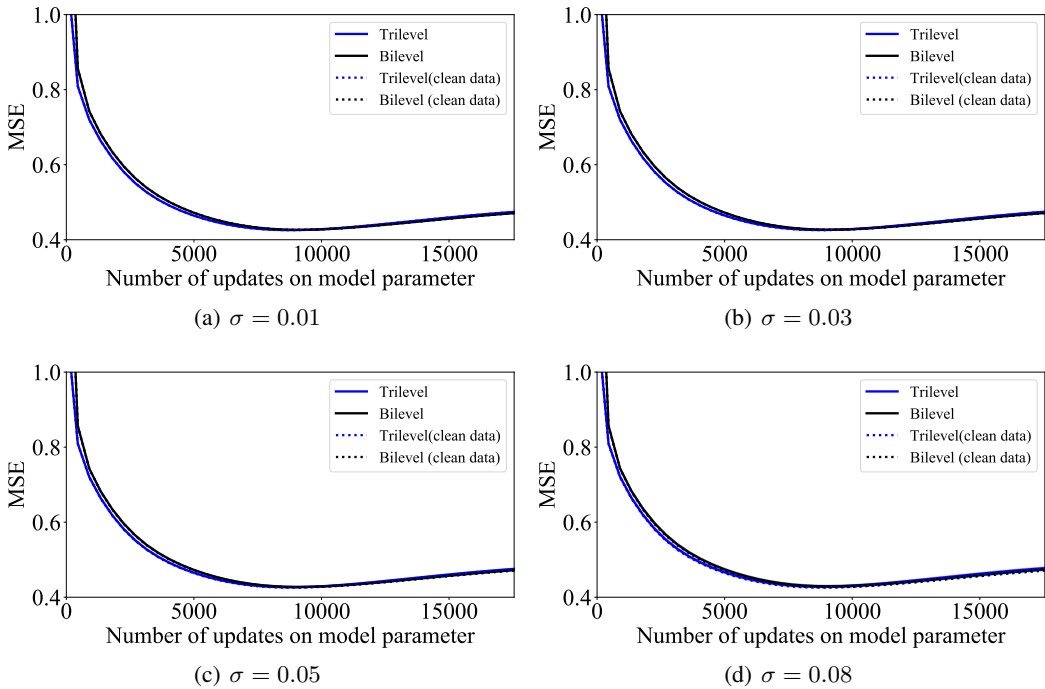

Figure 18: MSE of test data with Gaussian noise with a standard deviation of $\sigma$ on Boston house-prices dataset.

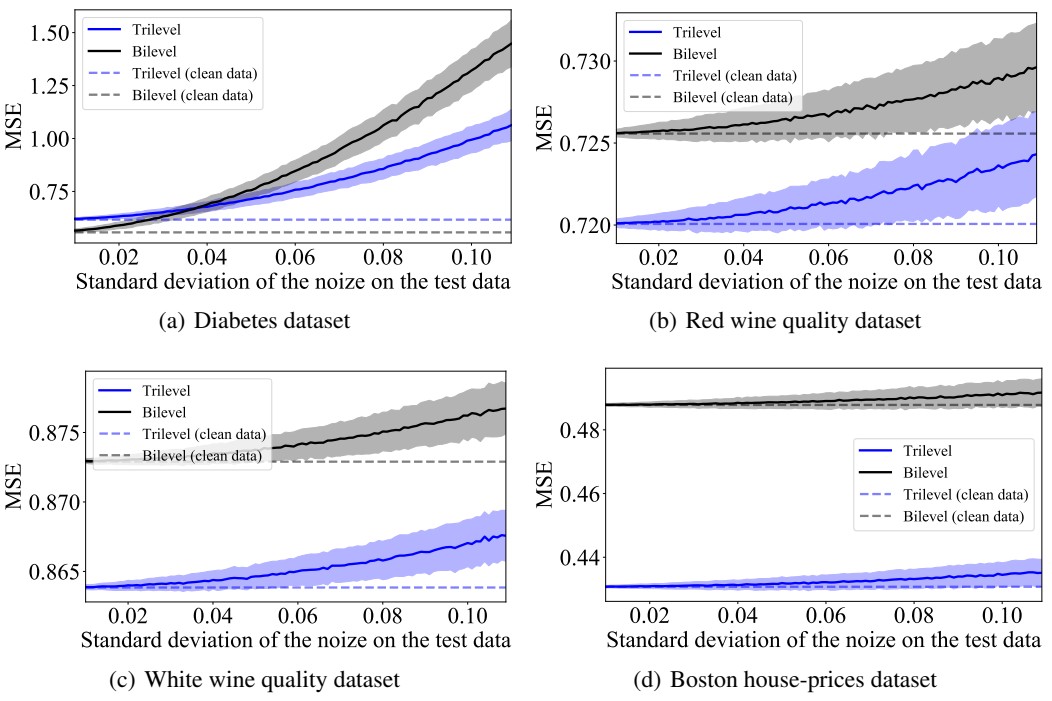

(a) Diabetes dataset

(b) Red wine quality dataset

(c) White wine quality dataset

(d) Boston house-prices dataset

Figure 19: MSE of test data with Gaussian noise, using early-stopped parameters for prediction.

Table 2: MSE of test data with Gaussian noise with standard deviation of 0.08, using early-stopped parameters for prediction. The better values are shown in bold face.

|  | diabetes | Boston | wine (red) | wine (white) |
|---|---|---|---|---|
| Trilevel | **0.8601 ± 0.0479** | **0.4333 ± 0.0032** | **0.7223 ± 0.0019** | **0.8659 ± 0.0013** |
| Bilevel | 1.0573 ± 0.0720 | 0.4899 ± 0.0033 | 0.7277 ± 0.0019 | 0.8750 ± 0.0014 |

the MSE of the trilevel model was consistently lower than that of the bilevel model. Therefore, the trilevel model provides more robust parameters than the bilevel model in these settings of problems.