# OpenReview forum: "A Gradient Method for Multilevel Optimization"
_NeurIPS.cc/2021/Conference — NeurIPS 2021 Poster_

### Official Review · Reviewer_cJPD · 2021-06-26

**Rating:** 7
**Confidence:** 4

**Summary:**

A gradient method for multilevel optimization is proposed with convergence guarantee demonstrated by both theoretical analysis and numeric experiments. This is initial theoretical work in nonconvex multilevel optimization.

**Ethical Concerns:**

I think this work has no ethical concerns.


**Limitations And Societal Impact:**

Both the authors and I agree that the major limitation is large gradient computation complexity for large n, so the authors left cheaper approximate gradient computation for future research.

Another limitation I think is the lack of overall computation complexity required to achieve epsilon-accuracy, which looks not hard to derive from the theorems in this paper.

Both the authors and I do not think this theoretical work could have any potential negative societal impact.

**Main Review:**

Originality: This study focuses on theoretical study of multilevel optimization that has very few existing works. The advantage over these existing works is clear, i.e., this work has both nonconvex theoretical guarantee and experimental evidence. The extension from bilevel optimization methods and its convergence proof are non-trivial. Related work seems to be adequately cited.

Quality: The proposed gradient method has convergence guarantee which is supported by both technically sound theorems and experiments. The weakness, i.e., large complexity under large n, is also supported by Theorem 6. The work is complete. However, it would be more convincing if you add analysis of convergence rate or computation complexity to achieve epsilon accuracy, and add numeric comparison with existing multilevel optimization algorithms (such as [19, 21]).

Clarity: This paper is clearly written and well organized. Some points may need more explanation which I will list later.

Significance: The results are important initial steps into the theoretical research of multilevel optimization, yet there are still many works for future research. I think the theoretical results are likely to be used and surpassed by future theoretical studies of multilevel optimization, and this gradient method could be applied to multilevel optimization applications of small n. (e.g. n=3).

Some questions and advice:

(1) You said your Assumption 1 is extended from [11], but I saw these assumptions in [12] not [11].

(2) In the equation right below eq. (5), I think $\Phi_i^{t_i}$, $t_i$, $T_i$ should be $\Phi_j^{t_j}$, $t_j$, $T_j$ respectively, yes?

(3) At the end of your introduction, you said “For example, if it is extended recursively using T steepest-descent sequential updates at each level, the problem size can be increased with $T^n$ variables. On the other hand, our formulation has polynomial-size $nT$ variables. ”
Could you explain how your method differs from “T steepest-descent sequential updates at each level” that uses $T^n$ variables? How did you decrease to $nT$ variables?

(4) In Theorem 4, do you mean the “optimal function value”?

(5) Typo: In line 2 of Algorithm 1: “for i:=2,…,n”.

(6) In Theorem 6, how are p and q defined? What objective function-related variables or hyperparameters affect p and q. Also, do p and q<1 or >1? That I think makes much difference.

(7) Since $\widetilde{F}_1$ is $L$-smooth, convergence rate of $||\widetilde{F}_1||$ (or maybe distance to a stationary point) is perhaps available, and the work can be more comprehensive if the overall complexity (defined as iteration complexity $\times$ complexity per iteration in Theorem 6) to achieve $||\widetilde{F}_1||<\epsilon$ is provided. That might be added to supplementary due to the page limit.

(8) In your first experiment, it converges significantly faster and in a more direct path to the optimal point with smaller T2 and T3, how to explain? How do T2 and T3 affect your second experiment?

(9) In your first experiment, what is an update of $x_3$? Is it one implementation of line 5 of Algorithm 1 for i=3? Why do you use the number of updates of $x_3$?

(10) In your second experiment, I think it better to define $||\cdot||_{1^*}$ explicitly or cite the reference that has the definition.

(11) In your second experiment, is this optimization problem borrowed from or inspired by any other references, or designed by yourself? If former, it is better to add citations.

(12) It could be better to compare your gradient method with existing multilevel optimization algorithms in experiments, such as [19, 21].

**Time Spent Reviewing:**

About 4 hours

---

> ### Author Response · Authors · 2021-08-10
> **Thank you for supporting our paper and providing constructive comments. We answer your questions and comments below.**
>
> Q1: You said your Assumption 1 is extended from [11],
> but I saw these assumptions in [12] not [11].
>
> A1: You are correct. We should cite not [11] but [12] here.
> We will correct it in the next update.
>
> Q2: In the equation right below eq. (5),
> I think $\Phi_{i}^{t_{i}}$, $t_{i}$, $T_{i}$ should be $\Phi_{j}^{t_{j}}$, $t_{j}$, $T_{j}$ respectively, yes?
>
> A2: Yes. Thank you for pointing out our mistakes.
> We will correct them to $\Phi_{j}^{(t_{j})}$, $t_{j}$, and $T_{j}$.
>
> Q3: At the end of your introduction, you said
> "For example, if it is extended recursively using $T$ steepest-descent sequential updates at each level,
> the problem size can be increased with $T^{n}$ variables.
> On the other hand, our formulation has polynomial-size $n T$ variables."
> Could you explain how your method differs from
> "$T$ steepest-descent sequential updates at each level"
> that uses $T^{n}$ variables?
> How did you decrease to $n T$ variables?
>
> A3: Following the suggestion from Reviewer RnFj,
> we will add some explanation on an approximated problem to be solved by our algorithm
> (i.e., Problem (5) for the trilevel case) by assuming a simple setting,
> where we apply the steepest descent method for the lower-level problems with the same iteration number $T$ and stepsize $\alpha$ for all levels as follows:
> \begin{equation}
> \begin{alignedat}{2}
> \min_{x_{1} \in S_{1}, \\{x_{2}^{(t)}\\}, \\{x_{3}^{(t)}\\}}{}&
> 	f_{1}(x_{1}, x_{2}^{(T)}, x_{3}^{(T)})\\\\
> \text{s.t. }&
> 	x_{2}^{(t)} = x_{2}^{(t - 1)} - \alpha
> 	\nabla_{x_{2}} \tilde{F}\_{2}(x_{1}, x_{2}^{(t - 1)}, x_{3}^{(T)})&
> 	    \quad& (t = 1, \dots, T),\\\\
> & 	x_{3}^{(t)} = x_{3}^{(t - 1)} - \alpha
> 	\nabla_{x_{3}} \tilde{F}\_{3}(x_{1}, x_{2}^{(T)}, x_{3}^{(t - 1)})&
> 	    \quad& (t = 1, \dots, T).
> \end{alignedat}
> \end{equation}
> We will explain more on
> "if  it  is  extended  recursively using $T$ steepest-descent sequential updates at each level,
> the problem size can be increased with $T^{n}$ variables"
> by showing the corresponding formulation:
> \begin{equation}
> \begin{alignedat}{2}
> \min_{x_{1} \in S_{1}, \\{x_{2}^{(t)}\\}, \\{x_{3}^{(t_{1}, t_{2})}\\}}{}&
> 	f_{1}(x_{1}, x_{2}^{(T)}, x_{3}^{(T, T)})\\\\
> \text{s.t. }&
> 	x_{2}^{(t)} = x_{2}^{(t - 1)} - \alpha \nabla_{x_{2}} \tilde{F}\_{2}(x_{1}, x_{2}^{(t - 1)}, x_{3}^{(T)})&
> 	    \quad& (t = 1, \dots, T),\\\\
> &	x_{3}^{(t_{1}, t_{2})} = x_{3}^{(t_{1}, t_{2} - 1)} - \alpha \nabla_{x_{3}} \tilde{F}\_{3}(x_{1}, x_{2}^{(t_{1})}, x_{3}^{(t_{1}, t_{2} - 1)})&
>         \quad& (t_{1} = 1, \dots, T; t_{2} = 1, \dots, T).
> \end{alignedat}
> \end{equation}
>
> Q4: In Theorem 4, do you mean the "optimal function value"?
>
> A4: Yes, "optimal value" in (a) means "optimal function value."
>
> Q5: Typo: In line 2 of Algorithm 1: "for $i := 2, \dots ,n$."
>
> A5: Thank you. We will correct it.
>
> Q6: In Theorem 6, how are $p$ and $q$ defined?
> What objective function-related variables or hyperparameters affect $p$ and $q$.
> Also, do $p$ and $q < 1$ or $> 1$?
> That I think makes much difference.
>
> A6: The constants $p$ and $q$ are employed to simplify nested big O notation $\mathrm{O}^{n}(\cdot)$ and
> they depend on the computational complexity of the gradient of the objective functions in each level.
> Also, $p$ and $q$ should be greater than 1.
> Since the gradient of the objective function in the lower level is used to calculate the gradient of the objective function in the upper level,
> the amount of calculation of the gradient at each level increases from the bottom to the top.
> From this, we can confirm $p, q > 1$.
>
> Q7: Since $\tilde{F}\_{1}$ is $L$-smooth, convergence rate of $\\|\tilde{F}\_{1}\\|$
> (or maybe distance to a stationary point) is perhaps available,
> and the work can be more comprehensive if the overall complexity
> (defined as iteration complexity * complexity per iteration in Theorem 6)
> to achieve $\\|\tilde{F}_{1}\\| < \epsilon$ is provided.
> That might be added to supplementary due to the page limit.
>
> A7: We can easily derive iteration complexity to achieve an $\epsilon$-stationary point.
> We will add (more rigorous description of) the following into the update:
> Let $G\colon \mathrm{int}(\mathrm{dom}(\tilde{F}\_{1})) \to \mathbb{R}^{d_{1}}$ be the gradient mapping [3, Definition 10.5]
> corresponding to $\tilde{F}\_{1}$, (the indicator function of) $S_{1}$, and the constant step size.
> Note that $\\|G(x)\\| = 0$ if and only if $x$ is a stationary point [3, Theorem 10.7].
> We have $\min_{s = 0}^{t} \\|G(x^{(s)})\\| \le O(1 / t)$ [3, Theorem 10.15 (c)].
> Hence, if $t \ge O(\epsilon^{2})$, $\\|G(x^{(s)})\\| \le \epsilon$ is achieved for some $s \le t$.
>
> Q8: In your first experiment,
> it converges significantly faster and in a more direct path to the optimal point
> with smaller $T_{2}$ and $T_{3}$, how to explain?
> How do $T_{2}$ and $T_{3}$ affect your second experiment?
>
> A8: In the first experiment, there is no complex relationship between variables at each level,
> and therefore, the objective function value at one level is not influenced significantly if variables at other levels is changed.
> In such a problem setting, by setting $T_{2}$ and $T_{3}$ to small values,
> we update $x_{1}$ many times and the generated sequence $\{x_{1}^{(t)}\}$ quickly converges to a stationary point.
> On the other hand, for example, if we set $T_{3}$ to a large value,
> $x_{3}$ is well optimized for some fixed $x_{1}$ and $x_{2}$, and hence, our whole algorithm may need more computation time until convergence.
> In the second experiment, the relationship between variables at each level is more complicated than in the first experiment.
> Setting $T_{2}$ or $T_{3}$ smaller in such a problem is not necessarily considered to be efficient
> because the optimization algorithm proceeds without fully approximating the optimality condition of $x_{i}$ at the $i$-th level.
> We will add this observation in the final version.
>
> Q9: In your first experiment, what is an update of $x_{3}$?
> Is it one implementation of line 5 of Algorithm 1 for $i = 3$?
> Why do you use the number of updates of $x_{3}$?
>
> A9: Yes, an update of $x_{3}$ means an execution of line 5 of Algorithm 1 for $i = 3$.
> We used the number of updates of $x_{3}$
> because updates of $x_{3}$ is the most inner iteration and the time required to update $x_{3}$ does not change when $T_{2}$ or $T_{3}$ changes
> and hence the number of updates of $x_{3}$ is proportional to the total computational time.
> By focusing on the number of updates of $x_{3}$,
> we can compare the time efficiency of the optimization algorithm.
> We will emphasize this in the final version.
>
> Q10: In your second experiment,
> I think it better to define $\\|\cdot\\|_{1^{\ast}}$ explicitly
> or cite the reference that has the definition.
>
> A10: We used $\\|x\\|\_{1^{\ast}} = \sum_{i = 1}^{k} \phi(x_{i}; \mu)$ with $\mu = 0.25$, where
> \begin{equation}
> \phi(x; \mu) =
> \begin{cases}
> x& (x > \mu),\\\\
> -\dfrac{x^{4}}{8 \mu^{3}} + \dfrac{3 x^{2}}{4 \mu^{2}} + \dfrac{3 \mu}{8}& (-\mu \le x \le \mu),\\\\
> -x& (x < -\mu)
> \end{cases}
> \end{equation}
> is a smoothed function of $|x|$ for $x \in \mathbb{R}$
> (Saheya et al.,
> Neural network based on systematically generated smoothing functions for absolute value equation,
> J. Appl. Math. Comput. 61 (2019) 533--558).
> We will add the reference in the update.
>
> Q11: In your second experiment,
> is this optimization problem borrowed from or inspired by any other references, or designed by yourself?
> If former, it is better to add citations.
>
> A11: It is inspired by (bilevel) hyperparameter optimization [12] and
> a type of adversarial learning called poisoning
> (Jagielski et al.,
> Manipulating machine learning: Poisoning attacks and countermeasures for regression learning,
> IEEE Symposium on Security and Privacy 2018, 19--35;
> Liu et al.,
> Min-max optimization without gradients: Convergence and applications to black-box evasion and poisoning attacks,
> ICML 2020, 6282--6293).
> We will cite these papers in the update.
>
> Q12: It could be better to compare your gradient method
> with existing multilevel optimization algorithms in experiments, such as [19, 21].
>
> A12: We are now executing numerical experiments to compare our proposed approach and evolutionary strategy [21].
> We will add results in the update.
> On the other hand, the approach based on fixed-point theory for trilevel optimization [19]
> requires the convexity of $f_{i}$ for all $i = 1, \dots, n$ and some more assumptions,
> which is more restrictive than our proposed approach.
> To numerically compare the two approach appropriately, we have to find instances
> that satisfy the assumptions in our paper and those in [19] simultaneously.
> After finding such instances, we will numerically compare our approach and [19].

---

> > ### Comment · Reviewer_cJPD · 2021-08-23
> > **Reviewer cJPD's 2nd comments**
> >
> > The authors well addresses my other questions except Q3.
> >
> > In the first equation in your reply to my Q3, the computation of $x_2^{(t)}$ requires $x_3^{(T)}$ while the computation of $x_3^{(t)}$ requires $x_2^{(T)}$. It seems that we need to solve a complicated equation about {$x_2^{(t)}, x_3^{(t)}$}$_t$, which is much more difficult than simple iteration. Should we remove $x_3^{(T)}$ in the definition of $x_2^{(t)}$, since $\widetilde{F}_i$ in your paper only takes inputs $x_1\ldots x_i$?
> >
> > Thank you for your response.
> >
> >
> > After reading the other two reviewers' comments and the authors' reply to them, I would like to say that I finally understand the algorithm and boosted my understanding by checking the proof of Theorem 5 (Gradient formula). The proof looks correct to me. Also, even if there is no experiment, I think the novel theoretical finding is also enough to be a Neurips paper. **I keep my original rating=7.**
> >
> > However, I think the authors still need to slightly edit the paper to explain the unclear points we pointed out, such as adding a 3-level instantiation of the algorithm, which the authors have promised to do.

---

> > > ### Author Response · Authors · 2021-08-25
> > > **Additional descripition about the procedure of Algorithm 1 and the trilevel instantiation**
> > >
> > > Thank you for asking the relation between updates of $x_{2}^{(t)}$ and $x_{3}^{(t)}$.
> > >
> > > First, in the definition of $x_{2}^{(t)}$, we need to include $x_{3}^{(T)}$
> > > because the objective value $f_{2}(x_{1}, x_{2}^{(t)}, x_{3})$ will change when $x_{3}$ changes.
> > >
> > > Second, we do not need to solve such a complicated equation.
> > > The problem we wrote as A3 in the above is equivalently written as Problem (6),
> > > which appears in the paper,
> > > minimizing the objective function $\tilde{F}\_{1}$.
> > > The calculation of $\nabla_{x_{1}} \tilde{F}\_{1}$ needs the evaluation of variables in a specific order as follows:
> > > $x_{3}^{(1)} \rightarrow \dots \rightarrow x_{3}^{(T)} \rightarrow x_{2}^{(1)}
> > > \rightarrow x_{3}^{(1)} \rightarrow \dots \rightarrow x_{3}^{(T)} \rightarrow x_{2}^{(2)}
> > > \rightarrow \dots \rightarrow x_{2}^{(T)}
> > > \rightarrow x_{3}^{(1)} \rightarrow \dots \rightarrow x_{3}^{(T)}$;
> > > that is, after one loop of updates of $x_{3}$,
> > > i.e., "$x_{3}^{(1)} \rightarrow \dots \rightarrow x_{3}^{(T)}$",
> > > $x_{2}$ is updated once and again we go back to the update loop of $x_{3}$.
> > > When the update loop of $x_{2}$ up to $x_{2}^{(T)}$ is over,
> > > we next update $x_{3}$ for $T$ times.
> > > At that point, the entire execution of Algorithm 1 is finished
> > > and we have $\nabla_{x_{1}} \tilde{F}\_{1}(x_{1})$ to update $x_{1}$.
> > > To find the optimal $x_{1}$,
> > > we next update $x_{1}$ once using $\nabla_{x_{1}} \tilde{F}\_{1}(x_{1})$,
> > > i.e., execute one iteration of a gradient-based algorithm,
> > > e.g., the projected gradient method, and then we go back to Algorithm 1.
> > > In the definitions of $x_{2}^{(t)}$ and $x_{3}^{(t)}$,
> > > the update process is not described entirely:
> > > only $x_{2}^{(T)}$ and $x_{3}^{(T)}$ computed with $x_{2}^{(T)}$ are written for simplicity.
> > > On the other hand, for each $t = 1, \dots, T - 1$,
> > > $x_{3}^{(T)}$ given $x_{2}^{(t)}$ is also computed in the order above,
> > > but that is not written explicitly in the definition of $x_{2}^{(t)}$.
> > > That might make it difficult to understand the procedure of computation.
> > > We will add these descriptions of trilevel instantiation of the algorithm in the final version.

---

> > > > ### Comment · Reviewer_cJPD · 2021-08-25
> > > > **Reviewer cJPD's 3rd comments**
> > > >
> > > > Thank you for your response.
> > > >
> > > > What you described here looks more like the second equation in your A3 to my Q3, which has T^n variables and T^n gradient descent steps. Is that right? How to explain that?
> > > >
> > > > Moreover, here you said the problem we wrote as A3 in the above is equivalently written as Problem (6), and your paper said Problem (6) is a reformulation of Problem (5), so I guess you mean the first equation in A3 is also equivalent to Problem (5), right? If so, in Problem (5), $x_j^{(t_i)}$ only relies on its higher-level variables $x_1, x_2^{T_i}, \ldots, x_{i-1}^{T_{i-1}}, x_i^{(t_i-1)}$. In contrast, the first equation in A3 let $x_i^{(t)}$ relies on all the variables including also $x_{i+1}^{(T)}$, etc. There seems to be a contradiction. How to explain that?
> > > >
> > > > Thank you.

---

> > > > > ### Author Response · Authors · 2021-08-26
> > > > > **Thank you for your important comments. We have noticed our important future work.**
> > > > >
> > > > > First, we have to say that there was confusion among the authors
> > > > > due to the complicated structure of multilevel optimization.
> > > > > We now have come to an agreement and we would like to share it to respond to your questions.
> > > > > In this reply, we call the upper and lower problems in the above two problems (A3.1) and (A3.2), respectively.
> > > > >
> > > > > We would like to answer your questions in a different order to make it easier to understand.
> > > > >
> > > > > Q13:
> > > > > Here you said the problem we wrote as A3 in the above is equivalently written as Problem (6),
> > > > > and your paper said Problem (6) is a reformulation of Problem (5),
> > > > > so I guess you mean the first equation in A3 is also equivalent to Problem (5), right?
> > > > > If so, in Problem (5), $x_{j}^{(t_{i})}$ only relies on its higher-level variables
> > > > > $x_{1}, x_{2}^{(T_{2})}, \dots, x_{i - 1}^{(T_{i - 1})}, x_{i}^{(t_{i} - 1)}$.
> > > > > In contrast, the first equation in A3 let $x_{i}^{t}$ relies on all the variables
> > > > > including also $x_{i + 1}^{(T)}$, etc.
> > > > > There seems to be a contradiction. How to explain that?
> > > > >
> > > > > A13:
> > > > > As you pointed out before, we should not put $x_{3}^{(T)}$ in the equation of $x_{2}^{(t)}$
> > > > > because $x_{3}^{(t)}$ does not give any influence on the computation of $\nabla_{x_{2}} \tilde{F}\_{2}$.
> > > > > Hence, we should have written (A3.1) as
> > > > > \begin{equation}
> > > > > \begin{alignedat}{2}
> > > > > \min_{x_{1} \in S_{1}, \\{x_{2}^{(t)}\\}, \\{x_{3}^{(t)}\\}}{}&
> > > > > 	f_{1}(x_{1}, x_{2}^{(T)}, x_{3}^{(T)})\\\\
> > > > > \text{s.t. }&
> > > > > 	x_{2}^{(t)}
> > > > > 	= x_{2}^{(t - 1)} - \alpha \nabla_{x_{2}} \tilde{F}\_{2}(x_{1}, x_{2}^{(t - 1)}),&
> > > > > 	    \quad& (t = 1, \dots, T),\\\\
> > > > > & 	x_{3}^{(t)}
> > > > >     = x_{3}^{(t - 1)} - \alpha \nabla_{x_{3}} \tilde{F}\_{3}(x_{1}, x_{2}^{(T)}, x_{3}^{(t - 1)})&
> > > > >         \quad& (t = 1, \dots, T)
> > > > > \end{alignedat}
> > > > > \end{equation}
> > > > > and (A3.2) as
> > > > > \begin{equation}
> > > > > \begin{alignedat}{2}
> > > > > \min_{x_{1} \in S_{1}, \\{x_{2}^{(t)}\\}, \\{x_{3}^{(t_{1}, t_{2})}\\}}{}&
> > > > > 	f_{1}(x_{1}, x_{2}^{(T)}, x_{3}^{(T, T)})\\\\
> > > > > \text{s.t. }&
> > > > > 	x_{2}^{(t)}
> > > > > 	= x_{2}^{(t - 1)} - \alpha \nabla_{x_{2}} \tilde{F}\_{2}(x_{1}, x_{2}^{(t - 1)})&
> > > > > 	    \quad& (t = 1, \dots, T),\\\\
> > > > > &	x_{3}^{(t_{1}, t_{2})}
> > > > >     = x_{3}^{(t_{1}, t_{2} - 1)}
> > > > >     - \alpha \nabla_{x_{3}} \tilde{F}\_{3}(x_{1}, x_{2}^{(t_{1})}, x_{3}^{(t_{1}, t_{2} - 1)})&
> > > > >         \quad& (t_{1} = 1, \dots, T; t_{2} = 1, \dots, T).
> > > > > \end{alignedat}
> > > > > \end{equation}
> > > > > We think there is no contradiction any more.
> > > > >
> > > > > In the following, we call these corrected problems above (A13.1), (A13.2), respectively.
> > > > >
> > > > > Q14:
> > > > > What you described here looks more like the second equation in your A3 to my Q3,
> > > > > which has $T^{n}$ variables and $T^{n}$ gradient descent steps.
> > > > > Is that right? How to explain that?
> > > > >
> > > > > A14:
> > > > > Based on your comments, we examined how to solve Problem (A13.2).
> > > > > As a result, we have noticed that our Algorithm 1 solves not only Problem (A13.1) but also (A13.2).
> > > > > Particularly, when we solve Problem (A13.1) using a gradient-based method based on Algorithm 1,
> > > > > by keeping the intermediate value of each variable, we can find a solution for Problem (A13.2).
> > > > > Therefore, we should not claim that we do not solve Problem (A13.2) but do (A13.1),
> > > > > and hence, we will remove the claim on the size of reformulation in Abstract and Section 1 in the update.
> > > > > Instead, we would like to include two reformulations,
> > > > > Problems (A13.1) and (A13.2) (or more rigorous descriptions),
> > > > > and discussion above into Supplementary materials.
> > > > > We also noticed that the construction of an algorithm
> > > > > that does not solve Problem (A13.2) but do only (A13.1)
> > > > > is left for our important future work.
> > > > > We would like to mention this future work in the final version.
> > > > >
> > > > > Although the important future work still remains,
> > > > > as far as we know, our algorithm is the first algorithm
> > > > > with theoretical guarantee and experimental verification
> > > > > for solving multilevel optimization with $n$ ($\ge 3$) level.
> > > > > We believe that a more theoretically and practically fast algorithm will be developed based on our paper.
> > > > >
> > > > > This discussion with you has significantly improved our paper.
> > > > > We are grateful to you for your kindness in spending a lot of time for review and discussion.

---

### Official Review · Reviewer_gn5o · 2021-07-16

**Rating:** 5
**Confidence:** 3

**Summary:**

The authors propose an approximate algorithm for multilevel optimization problems. The presented method can be considered an $n$-level extension of a related and previously published algorithm for bilevel optimization. It relies essentially on replacing optimal solutions in lower-level problems by respective $T$-step gradient descent approximations. That way, the original multilevel problem can be approximated in terms of an unconstrained problem depending only on top-level variables. After an an introduction and discussion of related work, the authors describe how the objective of the approximate problem can be differentiated with respect to the top-level variables. Moreover, assumptions are given under which the original problem admits an optimal solution. And it is shown that the optimal value of the approximate problem converges to a solution of the original problem in the limiting case $T\rightarrow\infty$. Finally, the authors report numerical results obtained through application of their method to two different trilevel optimization problems.

**Limitations And Societal Impact:**

Yes.

**Main Review:**

From my point of view, this is an interesting paper. The original problem formulation **(1)** is very general and should be applicable in many contexts. However, as the authors also state in their conclusion, their method does probably not scale well to multiple levels. Moreover, I think that the presented numerical examples are somehow narrow because not only the level is limited to $n=3$ but also the number of optimization variables is in both presented examples quite small.

 In a maybe different but related context, the technique of using algorithmic iterates to replace optimal solutions of lower-level problems has been applied frequently (see, e.g., **Algorithm unrolling: Interpretable, efficient deep learning for signal and image processing (Monga, Li, Eldar, 2021)** for a recent overview). I think, a few references to that body of work would have been appropriate.

In the numerical experiments section, the authors mention that they used automatic differentiation. I think it should be possible to minimize the approximate objective relying only on automatic differentiation. Maybe the authors could clarify whether they did so or used their own **Algorithm 1**, or a mixture of both.

Finally, I have to say that I did not check the proofs in the supplementary material in detail. Apart from that, I would say that this is a borderline case where I tend to reject due to the mentioned limitations.



**Time Spent Reviewing:**

5

---

> ### Author Response · Authors · 2021-08-10
> **Thank you for providing comments that will refine our paper. We address your detailed questions below.**
>
> Q1: From my point of view, this is an interesting paper.
> The original problem formulation (1) is very general and should be applicable in many contexts.
> However, as the authors also state in their conclusion,
> their method does probably not scale well to multiple levels.
> Moreover, I think that the presented numerical examples are somehow narrow
> because not only the level is limited to $n = 3$
> but also the number of optimization variables is in both presented examples quite small.
>
> A1: As you pointed out, our numerical experiments are for trilevel optimization and the sizes of datasets may be small.
> However, there is no algorithm with computational complexity analysis so far even for trilevel problems as far as we know, except for [19],
> which requires the convexity of $f_{i}$ for all $i = 1, \dots, n$ and some more assumptions
> and hence is more restrictive than our proposed approach.
> We think that larger-scaled multilevel problems may be solved
> by using a computer that has a large amount of memory and can perform linear algebra operations at high speed.
> As the performance of computers is improving steadily with the progress of technology,
> we believe that if a computer with better performance comes out in the future,
> it will be possible to handle $n \ge 4$ or a larger number of optimization variables.
> We also believe that this paper will lead to the development of more efficient algorithms for multilevel optimization in the future.
>
> Q2: In a maybe different but related context,
> the technique of using algorithmic iterates to replace optimal solutions of lower-level problems has been applied frequently
> (see, e.g., Algorithm unrolling: Interpretable, efficient deep learning for signal and image processing (Monga, Li, Eldar, 2021) for a recent overview).
> I think, a few references to that body of work would have been appropriate.
>
> A2: We appreciate your information about algorithm unrolling.
> Figure 1 in that paper may seem similar to our Algorithm 1,
> but we do not think that algorithm unrolling is a method to approximate the optimal solutions of the lower-level problems,
> as the method is learning a deep network corresponding to an iterative algorithm.
> As of now, we have not found advantages of using a deep network like the one in Figure 1 for updating variables based on our update formula,
> but we are willing to see if there is anything we can do in the future.
> If you have noticed the relationship between our approximation and algorithm unrolling,
> we would like to discuss about it here.
>
> Q3: In the numerical experiments section,
> the authors mention that they used automatic differentiation.
> I think it should be possible to minimize the approximate objective relying only on automatic differentiation.
> Maybe the authors could clarify whether they did so or used their own Algorithm 1, or a mixture of both.
>
> A3: We used a mixture of both.
> We used Algorithm 1 to calculate the gradient of $\tilde{F}\_{i}$,
> and used automatic differentiation to calculate the gradient of $\Phi\_{i}^{(t)}$.
> We will clearly write this in the final version.

---

### Official Review · Reviewer_RnFj · 2021-07-16

**Rating:** 7
**Confidence:** 2

**Summary:**

Authors propose a gradient based method to solve multilevel optimization problems. In addition , they propose an new estimator based on a tr-level optimization problem, which seems to perform statistical better than usual bilevel hyperparameter optimization.

**Main Review:**

The proposed estimator and algorithm seem very new to me. Theorems and experiments are convincing.
The main problem is the clarity of the paper: I spent quite a bit of time on the paper, and I did not understand the proposed algorithm.
I would not be able to implement the algorithm myself.

Could you instantiate the algorithm on a simple tri-level optimization example? (at least in appendix?) Maybe this would help to understand. (I spend some time on Figure 5 in appendix but it did not help me)
In particular I did not fully understand part 3.2 and part 4.1, I would recommend to try to smooth a little these parts

If authors manage to make the paper clearer, I will raise my score.

--------------------------------------------------------------------------------------------------

Comments after author response.

Authors have answered my questions, I think the paper is interesting and significantly new.
Thus I will raise my score to 7.

**Time Spent Reviewing:**

8h

---

> ### Author Response · Authors · 2021-08-10
> **Thank you for suggesting to include a simple example of trilevel optimization into this paper. We address your detailed questions below.**
>
> Q1: Could you instantiate the algorithm on a simple tri-level optimization example? (at least in appendix?)
> Maybe this would help to understand.
> (I spend some time on Figure 5 in appendix but it did not help me.)
> In particular I did not fully understand part 3.2 and part 4.1,
> I would recommend to try to smooth a little these parts.
>
> A1: We will add an approximated problem to be solved by our algorithm (i.e., problem (5) for the trilevel case) by assuming a simple setting,
> where we apply the steepest descent method for the lower-level problems with the same iteration number $T$ and stepsize $\alpha$
> for all levels as follows:
> \begin{equation}
> \begin{alignedat}{2}
> \min_{x_{1} \in S_{1}, \\{x_{2}^{(t)}\\}, \\{x_{3}^{(t)}\\}}{}&
> 	f_{1}(x_{1}, x_{2}^{(T)}, x_{3}^{(T)})\\\\
> \text{s.t. }&
> 	x_{2}^{(t)} = \Phi_{2}^{(t)}(x_{1}, x_{2}^{(t - 1)})
> 	:= x_{2}^{(t - 1)} - \alpha \nabla_{x_{2}} \tilde{F}\_{2}(x_{1}, x_{2}^{(t - 1)}, x_{3}^{(T)}),&
> 	    \quad& (t = 1, \dots, T),\\\\
> & 	x_{3}^{(t)} = \Phi_{2}^{(t)}(x_{1}, x_{2}^{(T)}, x_{3}^{(t - 1)})
>     := x_{3}^{(t - 1)} - \alpha \nabla_{x_{3}} \tilde{F}\_{3}(x_{1}, x_{2}^{(T)}, x_{3}^{(t - 1)})&
>         \quad& (t = 1, \dots, T).
> \end{alignedat}
> \end{equation}
> We replace $x_{2}^{(T)}$ in the objective function by
> $x_{2}^{(T - 1)} - \alpha \nabla_{x_{2}} \tilde{F}\_{2}(x_{1}, x_{2}^{(T - 1)}, x_{3}^{(T)})$
> and then replace recursively $x_{2}^{(t)}$  using $\\{x_{2}^{(t)}\\}\_{t = 0}^{T - 1}$.
> By applying the same procedure for $x_{3}^{(T)}$,
> we can reformulate the above problem to the one minimizing a function $\tilde{F}\_{1}(x_{1})$ under the constraint $x_{1} \in S_{1}$.
> Theorem 4 proves that the problem, $\min_{x_{1} \in S_{1}} \tilde{F}\_{1}(x_{1})$,
> converges to the trilevel optimization problem as $T \rightarrow \infty$.
> The gradient of $\tilde{F}\_{1}(x_{1})$ can be calculated by applying Algorithm 1 to this problem.
> Here, $\nabla_{x_{2}} \tilde{F}\_{2}(x_{1}, x_{2}^{(t - 1)}, x_{3}^{(T)})$ is the gradient of the objective function of
> $\min_{x_{2} \in \mathbb{R}^{d_{2}}} \tilde{F}\_{2}(x_{1}, x_{2})$,
> which is equivalent to the bilevel optimization problem as follows:
> \begin{equation}
> \begin{alignedat}{2}
> \min_{x_{2}^{(t)}, \\{x_{3}^{(t')}\\}}{}&
> 	f_{2}(x_{1}, x_{2}^{(t)}, x_{3}^{(T)})\\\\
> \text{s.t. }&
> 	x_{3}^{(t')} = x_{3}^{(t' - 1)} - \alpha \nabla_{x_{3}} \tilde{F}\_{3}(x_{1}, x_{2}^{(t)}, x_{3}^{(t' - 1)})&
>         \quad& (t' = 1, \dots, T).
> \end{alignedat}
> \end{equation}
> The explicit formula of $\nabla_{x_{1}} \tilde{F}\_{1}(x_{1})$ is given in Theorem 5 and hence we can compute it by using Algorithm 1.
> In Algorithm 1 to compute $\nabla_{x_{1}} \tilde{F}\_{1}(x_{1})$,
> the computation of $\nabla_{x_{2}} \tilde{F}\_{2}(x_{1}, x_{2})$ is also required.
> Its explicit formula is also given in Theorem 5 and hence we can compute it by using Algorithm 1.
> In Algorithm 1 to compute $\nabla_{x_{2}} \tilde{F}\_{1}(x_{1}, x_{2})$,
> the computation of $\nabla_{x_{3}} \tilde{F}\_{3}(x_{1}, x_{2}, x_{3})$ is also required.
> Since the third level problem is the lowest level problem, $\tilde{F}\_{3} = f_{3}$ holds and hence the computation of its gradient is easy.
> With these results, we can apply a gradient-based method (e.g., the projected gradient method) to minimize $\tilde{F}\_{1}(x_{1})$ using $\nabla \tilde{F}_{1}$.
> We will explain this in the final version.

---

> > ### Comment · Reviewer_RnFj · 2021-08-25
> > **Question on Algorithm 1**
> >
> > Thanks a lot for the detailed response.
> > I am currently trying to see if I understand better the algorithm, in particular, I am looking at Algo 1 in the paper.
> > As mentioned by another reviewer, it seems there is a typo in Algo 1.
> > Should all i from line 3 to 7 be k ? (if you remplace k by i in line 2 it seems that k is not defined then)

---

> > > ### Comment · Reviewer_RnFj · 2021-08-25
> > > **Other questions on Algorithm 1**
> > >
> > > Do we agree that $\Phi_i^{(t}$ do not only depend on $x_1, .., x_i$, but also on $x_{i+1}, ..., x_n$.
> > >
> > > From what I understand from Algorithm 1, you are solving the optimization "from top to bottom" from $2$ to $n$.
> > > Do you have any intuition on why we should do it in this sens? Not the other way around?

---

> > > > ### Author Response · Authors · 2021-08-25
> > > > **Relationship between our Algorithm 1 and automatic differentiation; Algorithm 1 can be regarded as forward mode automatic differentiation**
> > > >
> > > > Q2: Thanks a lot for the detailed response.
> > > > I am currently trying to see if I understand better the algorithm,
> > > > in particular, I am looking at Algo 1 in the paper.
> > > > As mentioned by another reviewer, it seems there is a typo in Algo 1.
> > > > Should all $i$ from line 3 to 7 be $k$?
> > > > (if you replace $k$ with $i$ in line 2 it seems that $k$ is not defined then.)
> > > >
> > > > A2: Yes, there are typos in Algorithm 1 as you and Reviewer cJPD pointed out.
> > > > We would like to change all $k$'s in lines 2 and 6 of Algorithm 1 to $i$'s
> > > > because they correspond to the index of the levels that are denoted by $i$ throughout the paper.
> > > >
> > > > Q3: Do we agree that $\Phi_{i}^{(t)}$ do not only depend on $x_{1}, \dots, x_{i}$,
> > > > but also on $x_{i + 1}, \dots, x_{n}$.
> > > > From what I understand from Algorithm 1,
> > > > you are solving the optimization "from top to bottom" from $2$ to $n$.
> > > > Do you have any intuition on why we should do it in this sense?
> > > > Not the other way around?
> > > >
> > > > A3: Our approach would correspond to the forward mode automatic differentiation.
> > > > There are two modes of automatic differentiation:
> > > > one is the forward mode, which follows a computational graph from top to bottom;
> > > > the other is the reverse mode, which does from bottom to top.
> > > > Therefore, to compute $\nabla \tilde{F}_{1}$,
> > > > there would be such two possible ways of automatic differentiation.
> > > > One method is our Algorithm 1 and it can be naturally regarded as the forward mode.
> > > > Meanwhile, there would be a possibility of another algorithm corresponding to the reverse mode,
> > > > and that is left for future work.

---

> > > > > ### Comment · Reviewer_RnFj · 2021-08-25
> > > > > **Algorithm 1**
> > > > >
> > > > > Thanks a lot for the clarifications.
> > > > > Maybe it worth writing a remark to talk about the links with usual forward differentiation.
> > > > > In particular, maybe you could write explicitely the cost of each step in the code between line 4 and 7 in Algorithm 1, and compare with the cost of the usual forward differentiation.

---

> > > > > > ### Author Response · Authors · 2021-08-25
> > > > > > **Thank you for the suggestion.**
> > > > > >
> > > > > > Thank you for the suggestion. According to your suggestion, we will discuss more on the links
> > > > > > with usual forward differentiation in the final version.
> > > > > >
> > > > > > Regarding the complexity (= cost) of Algorithm 1, we evaluated it in Theorem 6,
> > > > > > and its proof is included in Supplementary material A.4.

---

### Decision · Program_Chairs · 2021-09-27

**Decision:**

Accept (Poster)

**Comment:**

Reviewer anonymously agree that this paper proposes a novel and non-trivial method for an important problem. Some reviewers have also raised concerns regarding clarity that have been addressed by the authors during the rebuttal phrase. I urge the authors to incorporate their feedback in the camera ready.